# Optimization of a Lightweight Floating Offshore Wind Turbine with Water Ballast Motion Mitigation Technology

**William Ramsay** [1]**, Andrew Goupee** [1,*]  **, Christopher Allen** [2]**, Anthony Viselli** [2] **and Richard Kimball** [1]

1    Department of Mechanical Engineering, College of Engineering, University of Maine, Orono, ME 04469, USA
2    Advanced Structures and Composites Center, Orono, ME 04469, USA
*    Correspondence: agoupee91@maine.edu

**Abstract:** Floating offshore wind turbines are a promising technology for addressing energy needs by utilizing wind resources offshore. The current state of the art is based on heavy, expensive platforms to survive the ocean environment. Typical design techniques do not involve optimization because of the computationally expensive time domain solvers used to model motions and loads in the ocean environment. However, this design uses an efficient frequency domain solver with a genetic algorithm to rapidly optimize the design of a novel floating wind turbine concept. The concept utilizes a liquid ballast mass to mitigate motions on a lightweight post-tensioned concrete platform. The simple cruciform-shaped design of the platform made of post-tensioned concrete is less expensive than steel, reducing the raw material and manufacturing cost. The use of ballast water to behave as a tuned mass damper allows a smaller platform to achieve the same motions as a much larger platform, thus reducing the mass and cost. The optimization techniques applied with these design innovations resulted in a design with a levelized cost of energy of USD 0.0753/kWh, roughly half the cost of the current state of the art.

**Keywords:** floating offshore wind turbine; controls; tuned mass damper; optimization



## 1. Introduction

Modern society faces an existential dilemma. As industrialized countries support a modern lifestyle driven by consumerism, energy consumption continues to grow. Even amongst the highest energy users, the primary source continues to be non-renewable energy sources such as oil, coal, and natural gas [1]. Coupled with developing nations reliance on dirty fuel sources such as coal, a warming planet already seeing the effects of climate change, and increasing energy prices [2], the need for energy source diversification has never been stronger. Offshore wind power is a resource with strong potential to fill this need in the United States. In fact, while the total U.S. energy consumption is 13 quads/year [3], the total potential of offshore wind, accounting for losses and including conservative assumptions regarding technical, legal, regulatory, and social inhibiting factors, is still 2 quads/year [4]. With 58% of this potential in water depths requiring floating platforms, the potential for floating offshore wind technologies as part of the United States' power portfolio is strong.

The state of the art of floating offshore wind technology, however, is expensive. According to NREL, existing FOWT technologies have achieved a levelized cost of energy (LCOE) of USD 0.15–0.18/kWh, which is high compared to the USD 0.03–0.05/kWh for land-based turbines [5]. Much of this cost is from the steel used to make large and heavy platforms designed to keep the system as stable as possible, survive large sea storms, and maintain similar dynamics to onshore wind turbines. An arm of the Department of Energy, the Advanced Research Projects Agency—Energy (ARPA-E), which funds emerging but unproven technologies, identified floating offshore wind as a research area with significant potential because of the untapped but currently expensive power resource. To address

this cost difference, the ARPA-E Aerodynamic Turbines Lighter and Afloat with Nautical Technologies and Integrated Servo-control (ATLANTIS) program set out to generate "radically new FOWT designs with significantly lower mass/area; a new generation of computer tools to facilitate control co-design of the FOWTs; and generation of real-data from full and lab-scale experiments to validate the FOWT designs and computer tools" [6]. To bring floating offshore wind technology down to a competitive cost, the goal of this project is to design a floating offshore wind turbine concept with a USD 0.075/kWh or less LCOE. The current work fits into the first ATLANTIS program category. Building on the University of Maine's experience with post-tensioned concrete and a previous collaboration with NASA on tuned mass dampers utilizing ballast water to stabilize the platform, this project proposes a lightweight floating platform with significantly lower costs than the current designs. Additionally, in keeping with a controls co-design methodology, the platform is optimized for the lowest possible cost with the use of computationally efficient analysis tools.

*Proposed Design and Solution Method*

The three main types of floating offshore wind turbine platforms are spars, tension-leg platforms, and semi-submersibles. Spars achieve their stability with the restoring force created between the low center of gravity and the high center of buoyancy. However, they require deep drafts to achieve this stability, which also necessitates assembly offshore, increasing costs. Tension-leg platforms can be stable and light due to stability achieved from the tension in the mooring lines, but anchoring to the seabed is difficult, especially as wind turbine sizes increase. Finally, semi-submersible platforms achieve their stability from a large water plane area. Designs must be large enough to avoid typical wave period excitation ranges of 5–20 s, but since the period is inversely proportional to the water plane area, existing designs have been large, heavy, and therefore expensive [7].

The typical design process of a floating offshore wind turbine is performed sequentially, owing to the computationally intensive time domain simulations required. To satisfy the design requirements of the International Electrotechnical Commission, the combination of winds, waves, and currents for all of the design load cases requires thousands of simulations. As a result, platforms cannot be optimized with an analytical function due to the nonlinear design constraints. Furthermore, stochastic optimization techniques are infeasible when using all design load cases with time domain simulations due to the computational time required. In order to develop the novel cruciform platform concept with tuned mass damper (TMD) elements and simultaneously minimize the cost to meet the ARPE-E project goals, a novel optimization technique was developed.

Other projects have proposed solutions to floating offshore wind turbine optimization problems. Most focus on replacing time domain simulations in the optimization with various methods. In [8], a spar was developed by generating 12 feasible designs with a spreadsheet calculator, executing a frequency domain simulation to down-select the three best designs and then performing time domain simulations on the set to choose a finalized design. This approach is similar to the current work in the progression from hydrostatic calculations showing feasible designs to frequency domain simulations. However, with only 12 designs to choose from, there is no way to guarantee the search space is optimal, as one can accomplish by examining the statistics of repeated genetic algorithm (GA) runs. Additionally, with the manual manipulation involved in spreadsheet calculations, this limits the set of designs that could be considered, and subsequent redesigns would also be time-intensive.

Replacement of the time domain simulations was also proposed with the use of machine learning to develop a statistical model of a mooring system in [9]. A similar approach was taken in parts of the current work; to replace the wave loadings on the hull that are typically obtained from the potential flow solver WAMIT, a response surface model was developed. However, statistical methods based on training points from the full time domain simulation were deemed unsuitable. With the number of input variables

required for the floating platform problem presented here being six, the number of training points for a statistical model would have required too many time domain simulations to be practical.

A similar method to the present work was developed by [10], where they developed an analytical model to replace the time domain simulations. Their analytical model only considered a subset of the degrees of freedom, as the frequency domain simulation in this work does. In order to verify their analytical models, they were benchmarked against the time domain solver OpenFAST, similar to the present work. While [10] also used a damping device, their optimization only focused on the parameters for the damping device and not the platform itself to minimize the overall cost.

Overall, the existing literature has identified the need to optimize floating offshore wind platforms to help reduce costs compared with traditional sequential design methods. In particular, attempts at reducing the amount of time domain simulations have been proposed. However, the existing literature does not contain methods to optimize the entire system efficiently. For example, where tuned mass dampers have been studied, only the tuned mass damper was optimized for a pre-existing platform. Therefore, for a new generation of lighter platforms, there is a need for a comprehensive optimization technique combining the hydrostatic and dynamic models, including tuned mass damper physics with sufficient computational efficiency so as to practically generate designs.

The present work is based on the use of a TMD element to reduce the platform mass and a novel optimization approach to minimize the cost of the platform. Drawing from a 2018 proof-of-concept basin test of a 1:50 scale semi-submersible platform with TMDs utilizing water ballast, potential was seen for a platform concept taking advantage of the motion mitigation properties of the TMD [11]. A photo of the test is shown in Figure 1. Since semi-submersible designs already require significant amounts of ballast to float with much of their height underwater, the ballast water can be used by the TMD to stabilize the platform without adding mass. Furthermore, with the motion mitigation from the TMD, the wave periods do not need to be avoided so the waterplane area of the platform can be reduced, reducing the mass of the material used in the platform. The optimization assumed a TMD element-motivating water ballast as the effective mass of the TMD, similar to [11].

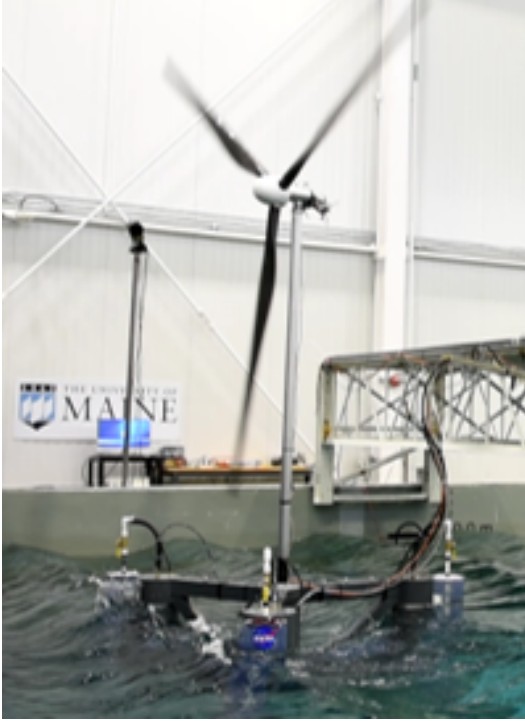

**Figure 1.** A photo from the 2018 model test.

The University of Maine has previous experience with post-tensioned concrete in the development of the VolturnUS semi-submersible floating offshore wind turbine platform [12]. Post-tensioned concrete is advantageous over steel in corrosion resistance, manufacturing cost, and material cost. With this in mind, the University of Maine developed a cruciform hull shape to be made of post-tensioned concrete, on which the current work would be based. The cruciform shape is easily constructed and allows room for ballast water and TMD equipment. The cruciform is shown in Figure 2. In keeping with industry trends toward larger turbines, the platform was designed around the IEA 15 MW reference turbine, a research turbine with power output consistent with state-of-the art and future industry turbines.

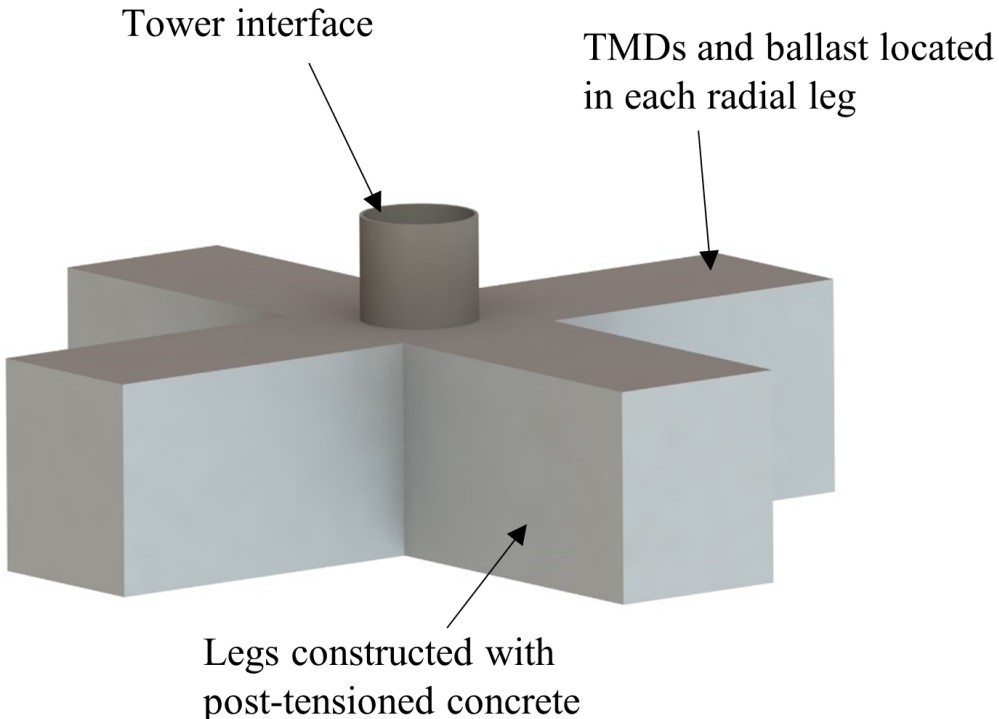

**Figure 2.** The cruciform hull concept.

Owing to the highly nonlinear constraints, a GA was chosen for the optimization architecture. A GA assesses the fitness of a given design based on the objective of the optimization, subject to constraints. The objective, minimization of the LCOE , was calculated based on a model developed by ARPA-E for the ATLANTIS program. Significant work was conducted on the development of the constraint functions. Similar to the requirements that would be set by a turbine's original equipment manufacturer, the typical values of horizontal and vertical acceleration and the pitch angle limits were set for an IEA 15-MW turbine. In addition, a model was required that accounted for the TMD and its travel limits. To capture these dynamic constraints, a frequency domain model was developed to save computational time over a time domain simulation. To generate the necessary inputs for the frequency domain model, a hydrostatic function was also developed. This model also outputs constraints related to geometric compatibility and initial stability. Since the hydrostatic constraints are essential to any design's suitability (a design that does not float is obviously not practical, for example), a staged constraint handling method was developed. When the hydrostatic constraints were violated, the GA skipped the execution of the frequency domain model. This saved significant computational time, because while the frequency domain model took at least 90 s to run, the hydrostatic model required less than 1 s.

The present work focuses on the optimization of the cruciform-type hull. In particular, the main developments were the input variable selection, integration of constraint functions

with the GA, development of a hydrostatic function to generate constraints and inputs to the frequency domain function, and control scheduling. The Materials and Methods section details the GA parameters, the staged constraint handling method, input variable selection, and details of the objective and constraint functions. Following that are the results, detailing the wind and wave conditions used, specifications of the IEA 15-MW turbine and the converged platform, simulation results for the platform, and convergence criteria for the GA. In all, the article describes the solution techniques required to optimize the novel cruciform hull design. This includes the numerical models developed specifically for this design and the wind and wave data necessary for their implementation. Tying together the numerical models is a genetic algorithm, with methods described to increase computational efficiency. Resulting from the optimization is a floating offshore wind platform with a significantly reduced cost compared with the state of the art.

## 2. Materials and Methods

After an initial platform concept was developed to demonstrate the potential for the ARPA-E ATLANTIS program, work began on development of the optimizer. The optimizer needed to produce results with enough fidelity to adequately describe the system while simultaneously being computationally efficient enough to allow 12,000 designs to be analyzed in a single optimization run. In summary, the typical analysis process of analyzing hydrostatic quantities and then using them as inputs in dynamic models was replaced by MATLAB functions executed sequentially in producing the fitness of a single design point. The details of the genetic algorithm optimizer and the MATLAB functions used to analyze the fitness of the designs are described in this chapter. Descriptions of the model use a coordinate system as shown in Figure 3.

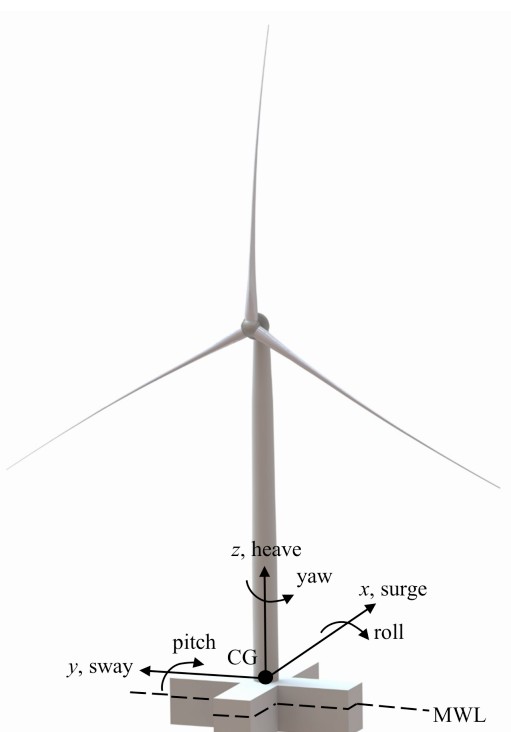

**Figure 3.** Coordinate system.

### 2.1. Genetic Algorithm and Constraint Handling

The optimization used a GA with tournament selection and niching as proposed by [13]. The present optimization follows the method in Section 3.4 of [14], which also used real coded variables defined as continuous rather than binary variables. The method aims to find the genes, or the specific values of input variables, that minimize a fitness function

composed of an objective and subject to constraints. The objective was minimization of the LCOE, and a number of constraints were imposed based on geometric feasibility, hydrostatic stability, and motion limits. The LCOE is defined as

$$LCOE = \frac{Total\ Lifetime\ Cost}{Total\ Lifetime\ Output} \tag{1}$$

Novel in this optimization effort was the use of a constraint function with a staged approach, whereby computationally inexpensive hydrostatic quantities were calculated first, and for those deemed infeasible, further calculations were not made. For those that passed the first round of constraints, more computationally expensive modeling was performed. The method of separating fitness and constraint functions so as to not penalize feasible design configurations was proposed in [13] and has been used extensively. In this optimization, there was further separation in the constraints based on first checking the hydrostatic and geometric criteria and skipping computationally intensive frequency domain calculations for infeasible designs from the first hydrostatic check. As such, the fitness of a given design was assigned to be

$$F(x) = \begin{cases} f(x), & \text{if } g_{HDF}(x) \ \& \ g_{FDF}(x) = 0 \\ f_{max} + g_{HDF}, & \text{if } g_{HDF}(x) > 0 \\ f_{max} + g_{FDF}, & \text{if } g_{HDF}(x) = 0 \end{cases} \tag{2}$$

where $x$ is a vector of the design parameters, $F(x)$ is the fitness, $f(x)$ is the objective function value, $g_{HDF}(x)$ is the hydrostatic function (HDF) which is less computationally expensive, $g_{FDF}(x)$ is the frequency domain function (FDF) which is more computationally expensive, and $f_{max}$ is the highest value of the objective function between two individuals in the tournament selection of the reproduction. Since this is a minimization problem, the lower the fitness $F(x)$, the better the design. In summary, the optimization is the minimization of LCOE ($f(x)$) subjected to hydrostatic constraints ($g_{HDF}$) and dynamic constraints ($g_{FDF}$). The GA is shown graphically in Figure 4.

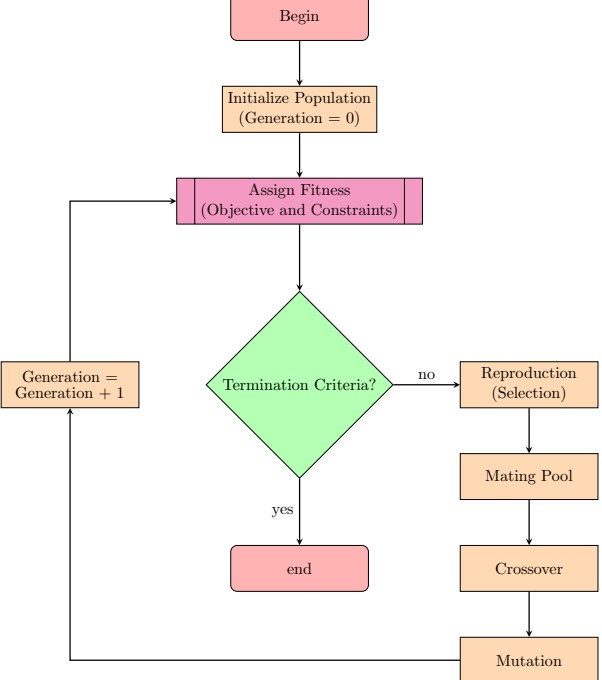

**Figure 4.** Flowchart of the GA.

The predefined process box for "Assign fitness" represents Equation (2), and the logic for determining the fitness value for one generation is depicted in Figure 5.

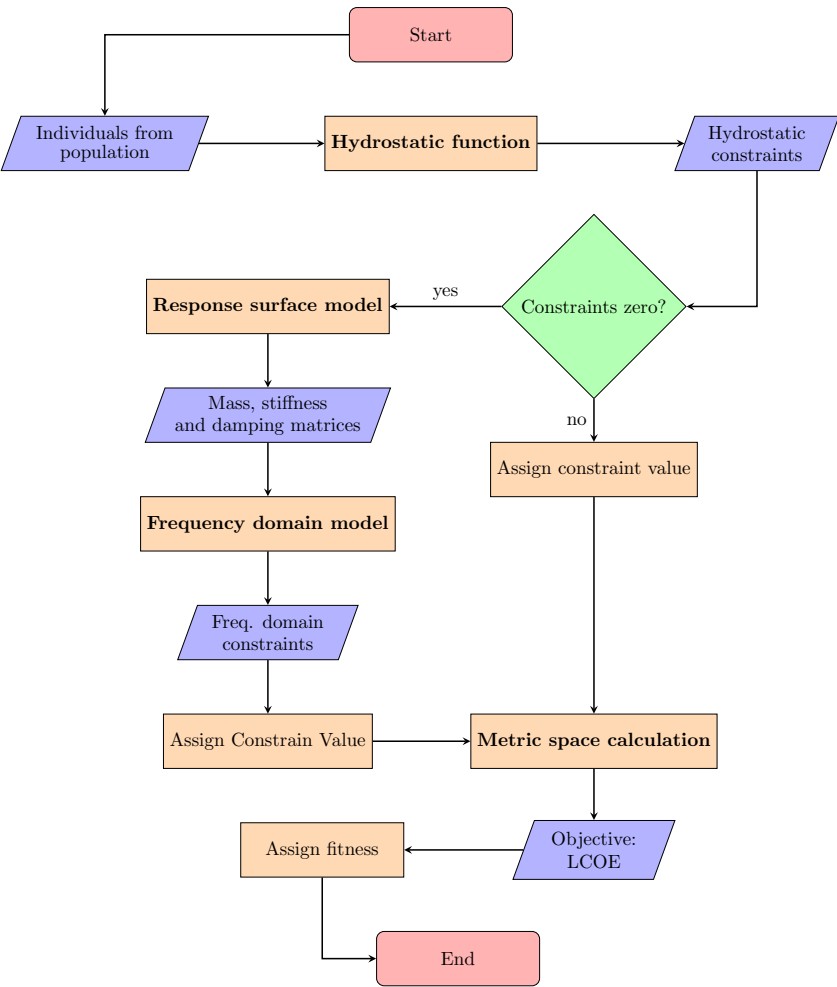

**Figure 5.** Flowchart of one iteration of the GA.

The bold text processes in Figure 5 are Matlab functions which are detailed in this chapter, and they comprised the majority of the research effort. The constraint values from the HDF and the FDF are also described.

### 2.1.1. Input Variables

The input variables were the following:

- $r$, the outer radius of the platform;
- $w$, the outer width of the platform;
- $d$, the draft of the platform;
- $h_p$, the displacement limit which is a bound on the travel of the TMD;
- $f$, the freeboard of the platform;
- $a$, the aspect ratio, which is the ratio between the inner length along the radius and the inner width of the platform.

The input variables are shown in Figure 6. $h_p$ is not included in this diagram because it describes the travel limit of the TMD.

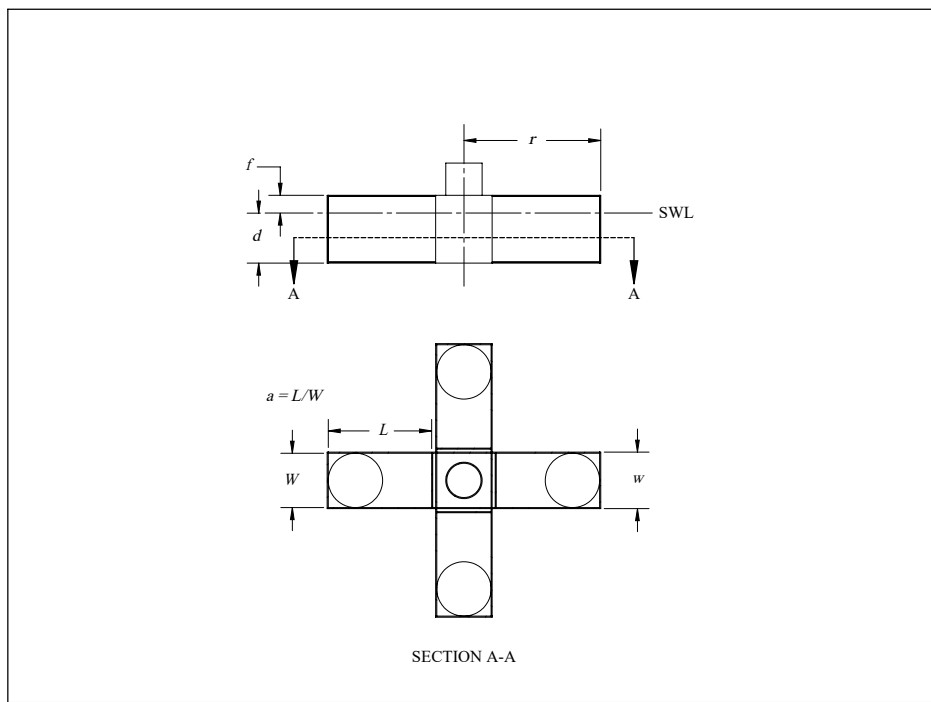

**Figure 6.** A diagram showing the definition of the input variables.

Selection of the input variables was based on the minimum number of variables to adequately affect the objective—minimization of the LCOE—and which had an effect on the constraints. The outer platform dimensions $r$, $w$, $d$, and $f$ influence the hydrostatics, static heel allowance, space for the ballast and TMD movement, dynamic response of the system, and the total mass of concrete, which is the main cost driver in the LCOE calculation. The displacement limit $h_p$ of the TMD affects the space available for the ballast, and importantly, the amount the TMD modeled in the FDF can move influences the dynamic performance. Finally, $a$ changes the space for the ballast water, in addition to the center of gravity of the ballast and the moment arm of the TMD.

The limits of the input variables are themselves geometric constraints and are as follows in Table 1.

**Table 1.** Input variables' ranges.

| Variable | Lower Limit | Upper Limit |
|:---:|:---:|:---:|
| $r$ (m) | 32.5 | 45 |
| $w$ (m) | 8 | 21 |
| $d$ (m) | 7.5 | 15 |
| $h_p$ (m) | 3 | 7 |
| $f$ (m) | 3 | 15 |
| $a$ | 1 | 2 |

The outer platform dimensions $r$, $w$, $d$, and $f$ were chosen based on an initial system design considering a set of reasonable designs in terms of initial hydrostatic stability and compatability with the IEA 15 tower and mass. The TMD travel range $h_p$ was chosen based on observing typical TMD motion extremes from the FDF and the upper limit such that there would be adequate space for the ballast water. The ballast tank aspect ratio $a$ tends toward filling the leg length, so it was set to be no less than 1, and the upper limit of 2 was near the full length of the leg for most width and radius combinations.

### 2.1.2. Constraints

The constraints were penalized differently based on the severity of their impact on platform feasibility. In particular, we have

$$
\begin{cases}
g_{HDF} = p_{HDF} \sum_{n=1}^{6} g_n + p_{FDF}, & \text{if } \sum_{n=1}^{6} g_n > 0 \\
g_{FDF} = p_{FDF} \sum_{n=7}^{10} g_n, & \text{if } g_{HDF} = 0
\end{cases}
\tag{3}
$$

where $g_{HDF}$ is the sum of the constraints calculated by the HDF, $g_{FDF}$ is the sum of the constraints calculated by the FDF, and $g_n$ is an individual constraint calculated by the HDF or FDF, of which there were 10 total. The penalties for each stage were $p_{HDF} = 1000$ and $p_{FDF} = 100$, and thus a more severe penalty on designs that fail the HDF constraints was assigned. If the HDF constraints failed, the FDF did not execute, and $p_{FDF}$ was added to the constraints to ensure the GA did not favor designs that just barely failed the HDF constraints. In Equation (3), on the first line, the sum of constraints reaches six, corresponding to the six constraints calculated by the HDF, and the four FDF constraints are not summed because the HDF constraints have failed (greater than zero). In line two of Equation (3), all the constraints are summed for both the HDF and FDF constraints because none of the HDF constraints are violated.

The constraints were normalized by a baseline value and by the number of constraints in their respective stage such that

$$
\begin{cases}
g_n = 0, & \text{if } x \geq x_b \\
g_n = \frac{x - x_b}{N x_b} & \text{if } x < x_b
\end{cases}
\tag{4}
$$

where $x_b$ is some baseline value, $x$ is the constraint quantity, and $N$ is the number of constraints in the stage. For some cases, the constraint value became infeasible when less than zero, in which case the constraint assigned was

$$
\begin{cases}
g_n = 0; & \text{if } x < 0 \\
g_n = \frac{-x}{N x_b} & \text{if } x \geq 0
\end{cases}
\tag{5}
$$

The constraints and their calculations were for the HDF and FDF:
HDF Constraints

- The hull is initially unstable: $g_1 = \dfrac{-GM}{N_h \cdot 16.44}$, where $GM$ is the metacentric height of the hull, and the baseline value of $GM = 16.44$ m is from an initial system design. This accounts for metacentric heights less than zero, which are obviously infeasible.

- The ballast water does not fit in the ballast chamber: $g_2 = \dfrac{y_{TMD} - y_{vac}}{N_h y_{TMD}}$, where $y_{TMD}$ is the travel limit of the TMD and $y_{vac}$ is the height of the vacant space in the ballast tank above the ballast water. If the required ballast mass with the TMD at the limit of its travel interferes with the top of the chamber, this constraint is non-zero.

- Negative ballast mass required: $g_3 = \dfrac{-m_b}{N_h \cdot 6.85 \times 10^6}$, where $m_b$ is the ballast mass in the hull and $6.85 \times 10^6$ kg is the ballast mass required from an initial system design. This accounts for situations where the buoyancy of the hull requires a negative ballast mass to reach the specified draft.

- Linear hydrostatics violated: $g_4 = \dfrac{-f_{min}}{N_h \cdot 3.79}$, where $f_{min}$ is the minimum freeboard under the rated thrust. This constraint becomes non-zero when the deck is just exposed to the waterline.

- Towout draft too large: $g_5 = \dfrac{d_{tow} - 10}{N_h \cdot 10}$, where $d_{tow}$ is the towout draft (the draft without ballast) and 10 m is the maximum draft allowable. This constraint ensures the hull does not sit too deep in the port.

- Ballast chamber does not fit: $g_6 = \dfrac{L_{bal} - L_{avl}}{N_h L_{avl}}$, where $L_{bal}$ is the length of the ballast chamber and $L_{avl}$ is the available space inside the hull along the radius for the ballast water. This accounts for situations where the combination of the aspect ratio and width is incompatible with the space available.

FDF Constraints

- The horizontal RNA acceleration limit is exceeded: $g_7 = \dfrac{a_{RNA,x} - 2.5}{N_f \cdot 2.5}$, where $a_{RNA,x}$ is the horizontal acceleration of the RNA and 2.5m/s$^2$ is a typical value set by a turbine OEM.

- The vertical RNA acceleration limit is exceeded: $g_8 = \dfrac{a_{RNA,z} - 2.0}{N_f \cdot 2.0}$, where $a_{RNA,z}$ is the vertical acceleration of the RNA and 2.0m/s$^2$ is a typical value set by a turbine OEM.

- The pitch angle limit is exceeded: $g_9 = \dfrac{\theta_p - 10}{N_f \cdot 10}$, where $\theta_p$ is the pitch angle of the tower and 10° is a typical value set by a turbine OEM.

- The TMD travel limit is exceeded: $g_{10} = y_{tmd}$, where $y_{tmd}$ is the maximum travel of the TMD. This constraint accounts for designs where there are no damper configurations (one period and varied damping ratios) that keep the TMD within the limits for all design load cases. See the section on the FDF for details on how the period and damping ratios were chosen.

2.1.3. Objective

The objective of the genetic algorithm was to minimize the LCOE. The objective function, as in Equation (2), was simply

$$f(x) = LCOE \tag{6}$$

Calculation of the objective was handled by the metric space calculation as shown in Figure 5. The metric space calculation was a model developed by ARPA-E for use by all projects in the ATLANTIS program, the details of which are described in the section on the metric space calculation.

*2.2. Hydrostatic Function*

The hydrostatic function is a computationally efficient MATLAB function to calculate the static stability and geometric compatibility constraints and generate inputs for the FDF. To allow geometry changes in MATLAB and to a Solidworks reference assembly, the cruciform hull was broken up into parallelepipeds parameterized to the overall dimensions of the system. The inputs are listed in Table 2.

**Table 2.** HDF inputs.

| Matlab Variable [1] | Description |
|---|---|
| $r, w, d, f, h_p, a$ | Optimizer variables as described in Table 1 |
| $h$ | Height, $f + d$ |
| $t$ | Nominal wall thickness, 0.3 m |
| $r_{ts}$ | Outer radius of tower support, 5 m |
| $h_s$ | Height of support above deck $15 - f$ |
| $n_{wall}$ | Number of additional walls for damage stability, 0 |
| $L_{bal}$ | Length of ballast tank, $a \cdot (w - 2t)$ |
| $r_p$ | Radius of TMD element |
| $A_0$ | Water plane area, $2wr + w(2r - w)$ |
| $V_0$ | Volume below waterline, $A_0 \cdot d$ |
| $F_b$ | Buoyant force, $gV_0 \cdot \rho_{ocean}$ |
| $I_{wp}$ | Waterplane area moment of inertia, $(2r - w)w^3/12 + w(2r)^3/12$ |
| $BM$ | Distance between center of buoyancy and metacentric height, $I_{wp}/V_0$ |
| $KB$ | Distance between keel and center of buoyancy, $d/2$ |
| $TMD_{lim,TMD}$ | Limit of TMD travel, $h_p - 0.5$ |
| $TMD_{lim,h20}$ | Limit of travel of water, $TMD_{lim,h20} \cdot \pi r_p^2/((w - 2t)L_{bal})$ |

[1] The variables under this heading are identically named to the variables in the MATLAB funcion, except where subscripts shown here are represented by underscores in the code.

The mass, KG, and mass moments of inertia were then calculated for each component and summed to obtain the overall system properties. Figure 7 shows the components of the platform, each of which is an element in the MATLAB function and Solidworks assembly. After the necessary system properties were calculated, the constraints were assigned.

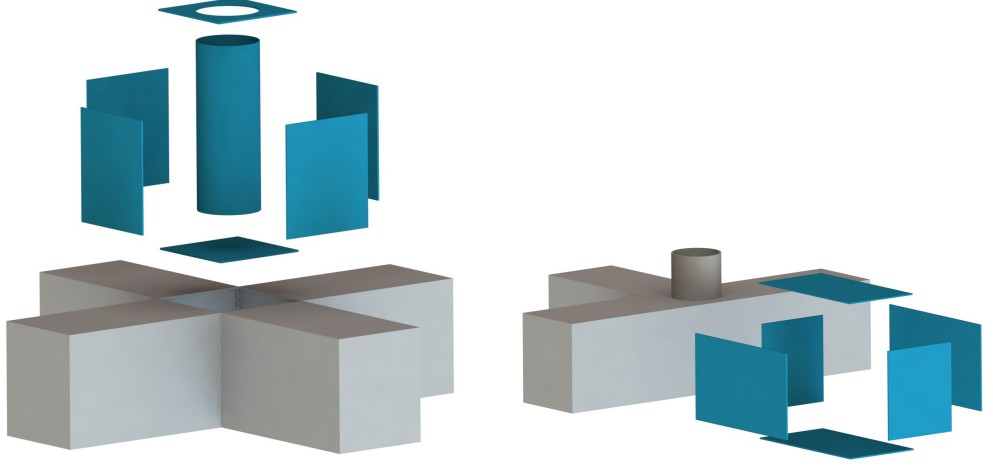

(**a**) An exploded view of the keystone       (**b**) An exploded view of one leg

**Figure 7.** Exploded views of the keystone (**a**) and one leg (**b**).

Before calculation of the constraints, the mass, center of gravity, and moments of inertia needed to be found. The masses of each component were obtained by the multiplication of the volume of each component and the concrete density and then summed to find the total mass as in

$$m = \sum_{i=1}^{n} \rho_c V_i \tag{7}$$

where the indices are $i$, the component, and $n$, the total number of components. $V$ is the volume of each hull component, and $\rho_c$ is the density of the steel-reinforced concrete. The volumes were parameterized to the system dimensions. For the tower, RNA, and blades of the IEA 15-MW turbine, the properties were from the publicly available reports from NREL [15,16].

The KG of each component was parameterized to the system dimensions and then summed to obtain the overall KG:

$$KG = \sum_{i=1}^{n} \frac{m_i \cdot KG_i}{m_i} \tag{8}$$

where *KG* is the distance from the keel to the center of gravity and *m* is the mass.

To obtain the mass moments of inertia *I* around the *x*, *y*, and *z* axes, the moments of inertia for each component were summed:

$$I = \sum_{i=1}^{n} I_i \tag{9}$$

In addition, the parallel axis theorem was applied to obtain the moments of inertia for each component:

$$I_i = I_{local} + m_i L^2 \tag{10}$$

where *L* is the distance between the *x*, *y*, or *z* axis passing through the component centroid and the hull centroid. Note that the ballast water was also modeled as a parallelepiped, and the free surface effects were ignored in calculating the static heel angle.

### 2.3. Frequency Domain Function

The frequency domain function is a two-dimensional, six-degree-of-freedom frequency domain dynamic response solver [17]. It considers wind and wave loading on the platform with sprung and damped lumped masses to represent the tuned mass damper system. A diagram of the model with degrees of freedom labeled is provided in Figure 8.

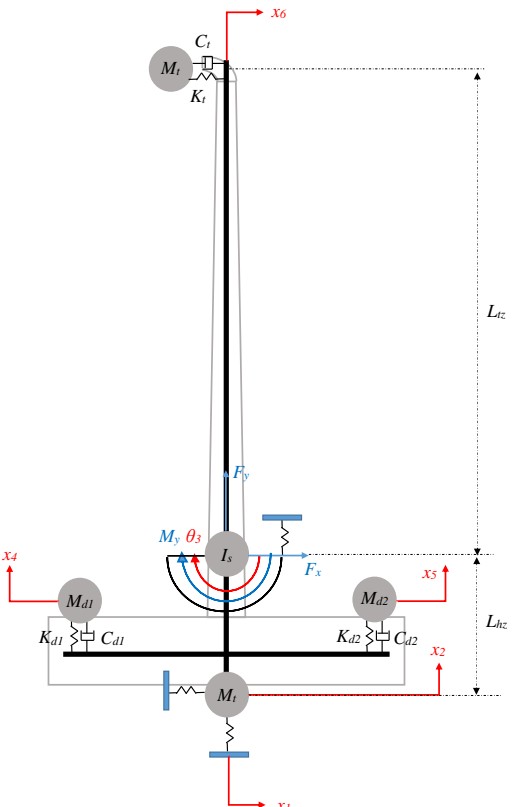

**Figure 8.** A diagram of the FDF model [17].

With the total mass, KG, and moment of inertia data calculated from the HDF, derivative quantities were used as the inputs for the FDF and as constraints. The key quantities input into the FDF are shown in Table 3.

**Table 3.** Frequency domain inputs.

| Matlab Variable | Description |
|---|---|
| $L_{wz}$ | Distance from the system CG to the waterline |
| $I_s$ | Mass moment of inertia in the pitch DOF about the center of gravity |
| $K_{11}$ | Mooring stiffness in the surge direction |
| $K_{33}$ | Heave stiffness |
| $z_{cg,tower}$ | Tower $z$ center of gravity |
| $M_{tower}$ | Mass of the tower |
| $z_{cg,hull}$ | Distance from CG of dry hull to system CG |
| $M_{hull}$ | Mass of the hull without ballast |
| $z_{cg,RNA}$ | RNA $z$ center of gravity |
| $M_{RNA}$ | Mass of the RNA |
| $Mp_{total}$ | Total ballast mass |
| $Mp_{xcg}$ | Ballast $x$ center of gravity |
| $Mp_{zcg}$ | Ballast $z$ center of gravity |
| $L_{tbz}$ | Distance from the system CG to the hull and tower interface |
| $h_{tank}$ | Inner height of the ballast tank |
| $w_{tank}$ | Inner width of the ballast tank |

To obtain the motion constraints, the outputs from Table 3 were passed into the computationally-efficient FDF. The FDF used wave forcing from WAMIT and the wind speed to aerodynamic loading transfer functions derived from OpenFAST and computes RAOs to output the response spectra and ultimate load information. For this optimization, the maximum acceleration of the RNA, maximum pitch angle, and maximum travel of the TMD were required to calculate the constraints.

2.3.1. Response Surface Model

Though shown as a separate function in Figure 5, the response surface model (RSM) was applied within the FDF. Typically, the hydrostatic stiffness coefficients, added mass and inertia coefficients, radiation damping coefficients, and wave excitation force and moments on a hull are obtained from a computationally intensive potential flow solver. However for the present work, an RSM was derived using inscribed central composite design points for the three input variables describing the hull below the waterline, radius, leg width, and draft. The design points used to train the RSM are shown in Figure 9.

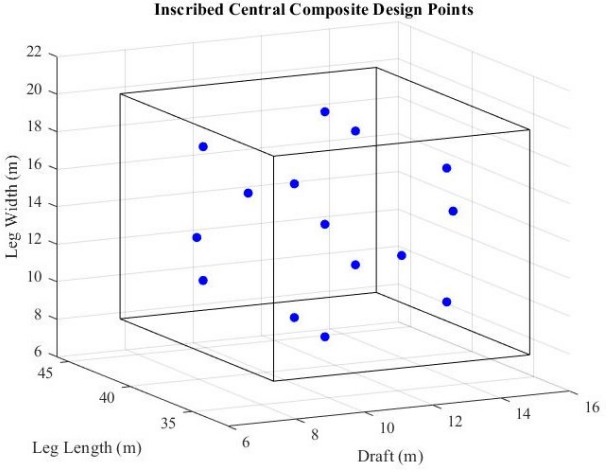

**Figure 9.** A graph showing the locations of the training points for the RSM.

Next, for each of the design points, a surface mesh was generated using MultiSurf [18], taking advantage of the symmetry in the two planes. For example, a mesh is shown in Figure 10.

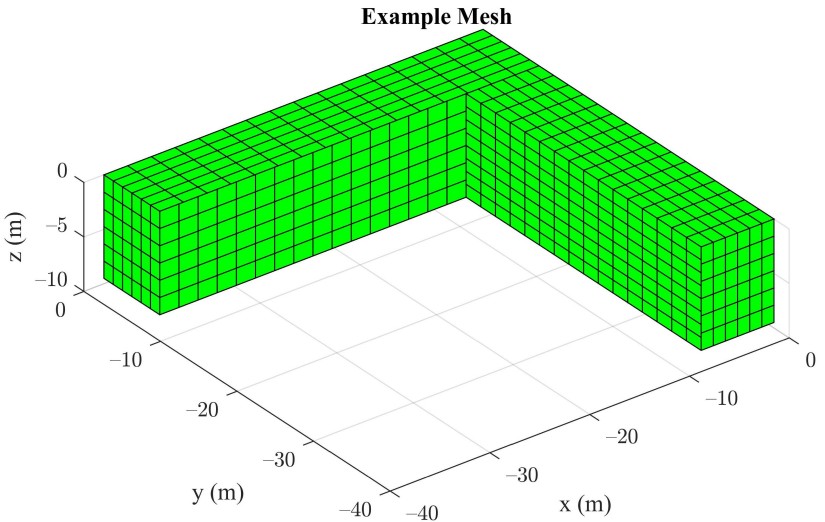

**Figure 10.** A graph showing a surface mesh of the platform below the waterline. Due to the symmetry in the two planes, only one quarter of the platform was generated.

Then, fully quadratic polynomial functions were fit to the hydrostatic coefficients in heave, roll, and pitch, as well as the added mass in all six degrees of freedom, the radiation damping coefficients in all six degrees of freedom, and the wave excitation forces and moments for all six degrees of freedom, wave periods, and wave headings in their real and complex components. To ensure an accurate fit, the results from WAMIT were compared to the polynomial function for a point not included in the inscribed central composite points. The WAMIT values versus the polynomial fit for $X_1$, the surge wave excitation force magnitude versus the period, are shown in Figure 11, indicating excellent agreement between the RSM and the WAMIT results. Each polynomial fit for the WAMIT quantities required was compared, with excellent agreement.

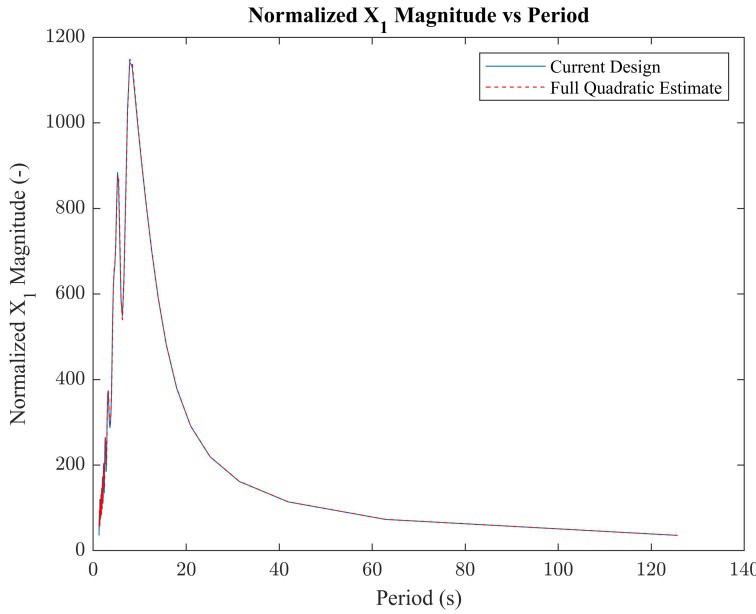

**Figure 11.** A graph comparing the $X_1$ values in terms of period from WAMIT with the polynomial fit.

### 2.3.2. Controller Scheduling

As detailed in [17], the FDF model output responses for a given sea state and range of a TMD period and damping ratios were not assigned as input variables to reduce the computation time. In order to assign FDF constraints, the response of the platform for a specific TMD period and damping value was needed. The FDF produced a matrix of values for each DLC case and each TMD configuration. The TMD was set to have a range of possible periods and damping values, with periods based on the bounds of typical ocean wave frequencies and the damping values within an assumed physically possible range. It was also assumed that any period could be set in the detailed design by the spring element. Thus, the output matrix had rows equal to the number of DLCs and columns equal to the number of periods considered times the number of damping values. As a result, the number of DLCs, periods, and damping ratios considered all added to the computation time. The period and damping ratio for the TMD were needed to obtain the dynamic response for each platform, but adding the damping ratio and period as variables to the optimization would have required a larger population in the GA, increasing the computation time. Furthermore, the best damping period varies by DLC, so there is not an obvious way to implement the damping as an input variable. Therefore, a control schedule was designed to minimize all platform motions while passing the constraints.

The controls over the TMD damping and period were scheduled with the assumption that a real control scheme would result in the minimum motion response of the platform. Since in a real embodiment, the spring would be fixed, but the damping could be changed along with the sea state on the scale of a few hours, and logic was implemented to choose the best damping ratio for each TMD period and DLC. There are multiple considerations in finding the best damping ratio. The first is that the TMD motion must stay within the travel limits inside the platform (constraint $g_{10}$). The second is that the RNA cannot exceed the acceleration and pitch angle limits (constraints $g_7, g_8,$ and $g_9$). The final consideration is that the motion should minimize the RNA accelerations and pitch angle. A weighted average of the platform constraints $g_7, g_8,$ and $g_9$ was used as the metric to minimize for the purpose of finding the best damper setting such that

$$\bar{R} = \sum_{n=7}^{9} \frac{r_{i,j}^{[n]}}{r_{max}^{[n]}} \tag{11}$$

where $\bar{R}$ is the weighted average of the platform motions, $r$ is the maximum platform motion for a given DLC, period, or damping ratio, the superscript $[n]$ corresponds to the platform constraint number (e.g., $r^{[7]}$, the maximum horizontal acceleration of the RNA, is used in the calculation of $g_7$), the subscript $i$ refers to the DLC, the subscript $j$ refers to the period and damping ratio combination, and the subscript $max$ refers to the limiting value as taken from the typical turbine OEM values used in the constraint calculation.

Based on a set range of DLCs, periods, and damping ratios, the FDF produced matrices of the maximum values for $r^{[6]}, r^{[7]}, r^{[8]},$ and $r^{[9]}$. For example, the TMD limits are in the form of Table 4. The limit of TMD travel varies based on platform geometry, and an example value of $r_{max}^{[6]} = 5.0$ m is used here. The values that passed are highlighted in green, and the values that failed are highlighted in red.

**Table 4.** Format of TMD motion matrix.

| DLC | $T_1$ | | | $T_2$ | | |
|---|---|---|---|---|---|---|
| | $\zeta_1$ | $\zeta_2$ | $\zeta_3$ | $\zeta_1$ | $\zeta_2$ | $\zeta_3$ |
| $DLC_1$ | 2.0 | 3.0 | 4.0 | 6.0 | 5.5 | 5.1 |
| $DLC_2$ | 5.5 | 4.0 | 4.5 | 5.5 | 4.0 | 6.0 |

Since a design whose TMD travel would exceed the physical space available is not feasible, the TMD travel is a factor in deciding the period and damping ratios. $r^{[7]}$,

$r^{[8]}$, and $r^{[9]}$. The damping ratio for each DLC is set based on the following logic: if all damping ratios pass as in $(T_1, DLC_1)$, then the chosen damping ratio is based on the best weighted average, calculated by Equation (11). For the case where at least one index fails but more than one passes, such as with $(T_1, DLC_2)$, then the chosen $\zeta$ is based on the lowest weighted average of those that pass. When only one $\zeta$ passes, such as with $(T_2, DLC_2)$, then that is the chosen $\zeta$. In the case of $(T_2, DLC_1)$, where no combinations pass, $\zeta$ is chosen such that $r^{[6]}$ is minimized. By applying this logic to matrices for $r^{[7]}$, $r^{[8]}$, and $r^{[9]}$, we might obtain examples such as those shown in Tables 5–8.

**Table 5.** Example RNA horizontal acceleration $r^{[7]}$.

| DLC | $T_1$ | | | $T_2$ | | |
|---|---|---|---|---|---|---|
| | $\zeta_1$ | $\zeta_2$ | $\zeta_3$ | $\zeta_1$ | $\zeta_2$ | $\zeta_3$ |
| $DLC_1$ | 1.0 | 2.0 | 3.0 | 1.0 | 2.0 | 3.0 |
| $DLC_2$ | 1.0 | 2.0 | 3.0 | 1.0 | 2.0 | 3.0 |

**Table 6.** Example RNA vertical acceleration $r^{[8]}$.

| DLC | $T_1$ | | | $T_2$ | | |
|---|---|---|---|---|---|---|
| | $\zeta_1$ | $\zeta_2$ | $\zeta_3$ | $\zeta_1$ | $\zeta_2$ | $\zeta_3$ |
| $DLC_1$ | 1.0 | 2.0 | 3.0 | 1.0 | 2.0 | 3.0 |
| $DLC_2$ | 1.0 | 2.0 | 3.0 | 1.0 | 2.0 | 3.0 |

**Table 7.** Example pitch angle $r^{[9]}$.

| DLC | $T_1$ | | | $T_2$ | | |
|---|---|---|---|---|---|---|
| | $\zeta_1$ | $\zeta_2$ | $\zeta_3$ | $\zeta_1$ | $\zeta_2$ | $\zeta_3$ |
| $DLC_1$ | 8.0 | 9.0 | 10.5 | 8.0 | 9.0 | 10.5 |
| $DLC_2$ | 8.0 | 9.0 | 10.5 | 8.0 | 9.0 | 10.5 |

Note that the values used in Tables 5, 6 and 8 are only examples and not representative of a real system. In addition, recall that $r_{max}^{[7]} = 2.0\,\text{m/s}^2$, $r_{max}^{[8]} = 2.5\,\text{m/s}^2$, and $r_{max}^{[9]} = 10.0°$. The green highlighted cells passed both the TMD travel limits and the respective platform motion constraints, the orange values passed the platform motion constraints but failed the TMD travel limits, and the red values failed just the platform motion constraints or both the platform motion constraints and the TMD motion constraints. When applying the TMD schedule, the resulting damping ratios are shown in Table 8.

**Table 8.** Damping ratios.

| DLC | $T_1$ | $T_2$ |
|---|---|---|
| $DLC_1$ | $\zeta_1$ | $\zeta_3$ |
| $DLC_2$ | $\zeta_2$ | $\zeta_2$ |

$\zeta_1$ for $(T_1, DLC_1)$ was chosen because all TMD travel values were below the limit, and $\zeta_1$ resulted in the best weighted average for $r_{[7]}$, $r_{[8]}$, and $r_{[9]}$. For $(T_1, DLC_2)$, $\zeta_2$ was chosen because although $\zeta_1$ resulted in a lower weighted average for $r_{[7]}$, $r_{[8]}$, and $r_{[9]}$, the TMD travel was too great. $\zeta_3$ was the resulting choice for $(T_2, DLC_1)$ because all three values of TMD travel were too high, but $\zeta_3$ was the lowest. Finally, $\zeta_2$ was chosen for $(T_2, DLC_2)$ because it was the only value with a low enough TMD travel value.

2.3.3. Design Load Case Down-Selection

Only a subset of DLCs from the ABS "Global Peformance Analysis of Floating Offshore Wind Turbine Installations" [19] was included in the FDF. The load cases considered are shown in Table 9.

**Table 9.** Design load cases.

| Condition | DLC |
|---|---|
| Power production, normal sea state | 1.1 |
| Power production, extreme sea state | 1.6 |
| Parked, 50 years of wind and waves | 6.1 |

The DLCs were chosen to have the relevant cases that would result in the worst values for the FDF constraints under normal and storm conditions. Therefore, startup, shutdown, and damage stability cases were not simulated due to the need to minimize the computation time and the increase in complexity of the HDF model for damaged cases. A detailed design review that went through all of the DLCs was conducted after the optimization effort.

To further reduce the computational time, certain wind bins were not included in the FDF. To identify which wind bins could be neglected, the FDF constraints were recorded for each wind bin in DLC 1.1 and 1.6 across a range of design points in the search space. If a certain wind bin never resulted in the maximum value for $r_{[7]}$, $r_{[8]}$, or $r_{[9]}$ across all damping ratios and periods considered, then it was neglected in the optimization. Table 10 shows the wind bins considered for DLC 1.1 and 1.6. A complete description of the wind and wave environment can be found in the Results section.

**Table 10.** Wind bins for DLC 1.1 and 1.6.

| DLC | Wind Bins (m/s) |
|---|---|
| 1.1 | 10, 24 |
| 1.6 | 10, 12, 14, 16, 18, 20, 22, 24 |
| 6.1 | 50 years of wind and waves |

For the normal operational case DLC 1.1, the wind bin's near-rated speed and the maximum wind speed were necessary. For the extreme sea state operational case DLC 1.6, the wind speeds from near-rated to the maximum wind speed were all considered.

With the input variables input into the HDF, the necessary constraints and inputs for the FDF were generated. Then, the dynamic constraints were assigned, and all constraint values were known for a given configuration. The next step was to assign the objective value.

*2.4. Metric Space Calculation*

The ARPA-E ATLANTIS program compares the designs from a variety of projects, and so a model was developed to compare the costs of each project [20]. The calculation of the LCOE, taken from [20], is defined as

$$LCOE = \frac{FCR \cdot CapEx + OpEx}{AEP} \tag{12}$$

where *FCR* is the fixed charge rate (1/year), *CapEx* is the capital expenditures (USD), *OpEx* is the capital expenditures (USD/year), and *AEP* is the annual energy production (kWh). The LCOE has units of USD/kWh.

To calculate the CapEx, in [20], the cost of multiple materials was combined into an equivalent mass of steel of the platform by material multiplication factors. Specifically, from [20], we have

$$m_j = f_{tj}(1 + f_{mj} + f_{ij})m_{cj} \tag{13}$$

where the index $j$ refers to the wind turbine component, $m$ is the equivalent mass of the component, $f_t$ is the material factor, $f_m$ is the manufacturing factor, $f_i$ is the installation factor, and $m_c$ is the mass of the component [20]. The material factors are reproduced in Table 11, and the manufacturing and installation factors are shown in Table 12.

**Table 11.** Metric space material factors.

| Material | $f_t$ | UMaine Adjusted $f_t$ |
|:---:|:---:|:---:|
| Aluminum alloys | 4.0 | - |
| Brass (70Cu30Zn, annealed) | 1.1 | - |
| CFRP laminate (carbon fiber reinforced polymer) | 80.0 | - |
| Copper alloys | 1.5 | - |
| GFRP laminate (glass-fiber, reinforced plastic, or fiberglass) | 4.0 | - |
| Lead alloys | 0.6 | - |
| Nickel alloys | 3.0 | - |
| Pre-stressed concrete | 0.3 | 0.13 |
| Titanium alloys | 22.5 | - |
| Steel of reference to calculate $f_t$ factors | 1.0 | - |

**Table 12.** Metric space manufacturing and installation factors.

| Component | $f_m$ | $f_i$ |
|:---:|:---:|:---:|
| Rotor | 3.87 | 0.10 |
| Hub | 11.00 | 0.10 |
| Nacelle | 9.49 | 0.10 |
| Tower | 1.69 | 0.10 |
| Floating platform | 2.00 | 0.13 |
| Mooring system | 0.14 | 0.52 |
| Anchor system | 6.70 | 3.48 |

The hull in this optimization was constructed of pre-stressed concrete, and UMaine's experience with pre-stressed concrete justified the reduction of the material factor from 0.3 to 0.13. Specifically, the new material factor was proposed based upon cost estimating completed for the DOE Wind Energy Technology Office under UMaine-led contracts DE-EE0006713.0000 and DE-EE0005990.0000. UMaine obtained three independent material, construction, and assembly quotes for 6-MW concrete hulls for 500-MW farms. For simplicity in the calculation worksheet, a single material factor $f_t$ of 0.13 was selected to reflect the cost estimating data for materials, construction, and assembly for the material, and therefore $f_m$ and $f_i$ were not changed.

An additional change was made to the sum of the masses. The array $m_{cj}$ was composed of the rotor, hub, nacelle, tower, floating platform, mooring system, and anchor system masses. Although the TMD equipment was made of steel, it was added directly to the platform mass as

$$m_{c5} = m_{conc} + 4m_{TMD} \tag{14}$$

where $m_{platform}$ is the mass of the platform in concrete and $m_{TMD}$ is the mass of the TMD equipment in one leg of the platform.

Mechanical System Costs

Finally, an additional change was made to the metric space to include the costs of mechanical equipment. ATKINS Houston Offshore Engineering was contracted to develop a module to calculate the cost of mechanical equipment for the floating platform. Earlier in

the life cycle of the project, a different configuration of the TMD element was being considered, for which the mechanical costing model was developed. Although the configuration changed, the main sensitivity of the model involved the cost of pressure vessels and compressors, which were still present in the current configuration at similar pressures. While time constraints did not allow the development of a model specific to the current system, because of the similarity of the equipment, it was considered to be sufficiently accurate. Furthermore, it is important to note that the cost of the mechanical equipment does not exceed 0.54% of the entire system cost, so its contribution is small.

The inputs to the mechanical costing model that changed during the optimization were the leg length, width, and height, the ballast tank length, width, and height, the air reservoir length, width, and height, and the pressure required. To demonstrate their impact on the LCOE, each of these variables was varied over their possible range while holding the other variables constant. A plot of this is shown in Figure 12.

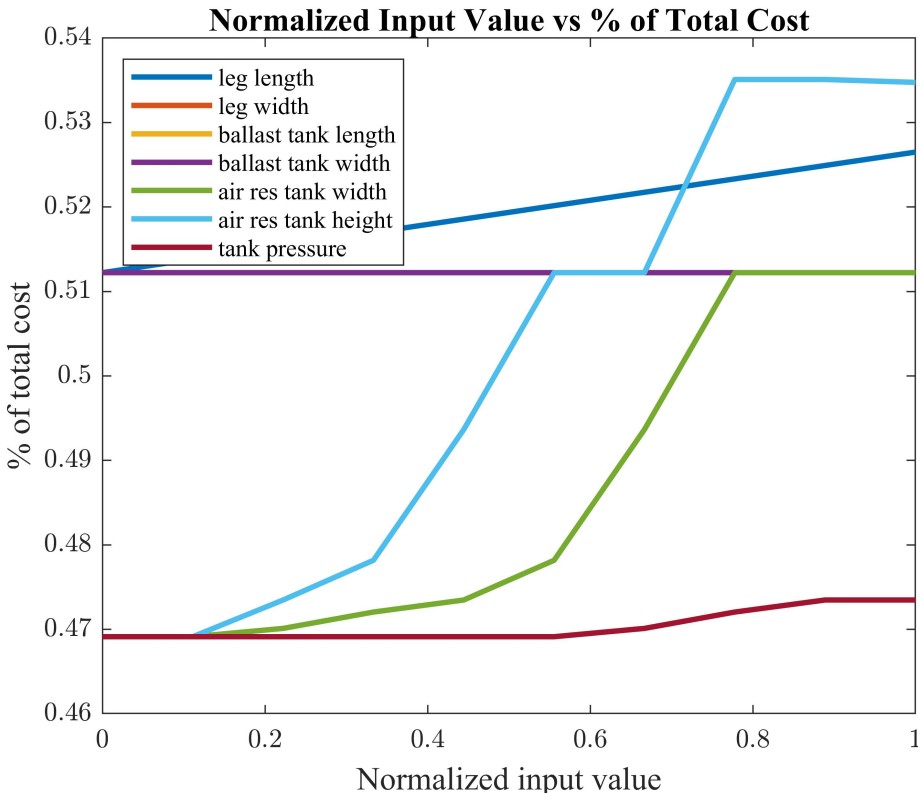

**Figure 12.** Graph of the percentage of the total system cost for each input variable.

As shown in Figure 12, the cost of the mechanical equipment was very small relative to the total system cost. It varied from 0.47% to 0.54% at most. Therefore, although it is not a perfect representation of the optimized system, it was included to capture the mechanical system cost trend.

## 3. Results

### 3.1. Optimized Platform Summary

The optimized platform used post-tensioned concrete in a simple cruciform shape in conjunction with damping devices in each radial leg, utilizing ballast water to reduce platform motion. The use of post-tensioned concrete reduced the manufacturing cost and material cost of the hull significantly. Furthermore, the addition of the damping devices allowed a smaller and lighter hull than traditional buoyancy-stabilized FOWT hull designs. Typically, designs such as semi-submersibles or barges achieve much of their rotational stiffness from the waterplane area moment of inertia. To gain the required area moment

of inertia, one may increase the area of the platform's cut waterplane section. However, this results in an undesirable increase in heave stiffness and produces minimal added pitch inertia, which can place the heave and pitch natural frequencies close to the wave energy range [21]. As such, it is general practice to achieve adequate pitch stiffness by increasing the distance of the waterplane area from the neutral axis, which can require a significant amount of structural framework to achieve. However, the addition of the damping devices allowed for the system's rigid body natural frequencies to lie within the wave excitation range, with the platform relying on the dampers to mitigate undesired resonant excitation. Finally, the platform was designed around the IEA 15-MW reference turbine, a theoretical turbine designed to represent the industry trend of larger capacity turbines. A rendering of the optimized platform design is shown in Figure 13.

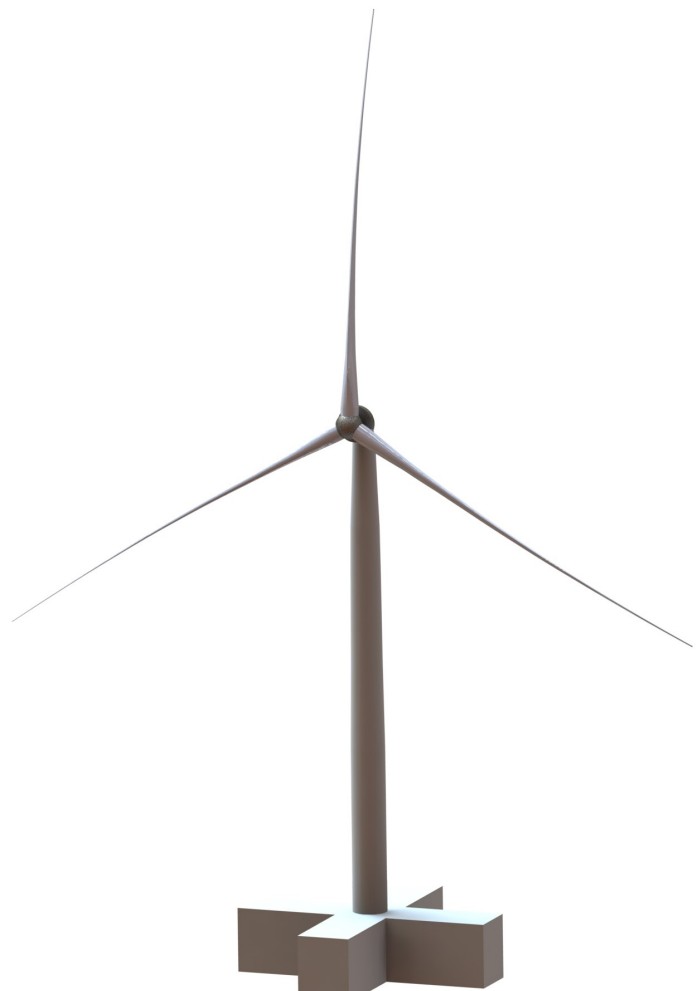

**Figure 13.** Rendering of the converged platform with the IEA 15-MW turbine.

Table 13 lists the mass of each component, the equivalent mass of the system in terms of the reference steel (see the metric space calculations), and each component's percentage of the equivalent steel mass. Current platform designs account for more than 50% of the equivalent mass of the entire system, according to ARPA-E analysis developed from [22]. The major advantage of this design is that the percentage of equivalent steel mass for the floating platform is roughly 15% of the total mass, allowing a significant reduction in overall cost.

**Table 13.** Mass and equivalent masses of platform components.

| Item | Actual Mass (kg) | Equivalent Steel Mass (kg) | Percentage of Equivalent Mass (%) |
|---|---|---|---|
| Rotor | 194,126 | 3,859,200 | 18.5 |
| Hub | 190,000 | 2,299,000 | 11.0 |
| Nacelle | 607,275 | 6,431,000 | 30.9 |
| Tower | 1,262,967 | 3,523,700 | 16.9 |
| Floating Platform | 7,905,400 | 3,216,700 | 15.4 |
| Mooring System | 140,040 | 232,470 | 1.12 |
| Anchor System | 114,000 | 1,274,520 | 6.12 |

The optimization effort using the genetic algorithm proved successful, with adequate computational efficiency. The staged constraint method coupled with the frequency domain function and parallel processing allowed for a relatively fast computational speed. The use of an engineering workstation laptop executed the optimization in between 1 and 2 days. Furthermore, a solution was found that met the cost targets and passed the constraints, reaching the goals of the ARPA-E project. Overall, ARPA-E set a cost target of USD 0.075/kWh, and the optimizer produced a platform design of USD 0.0753/kWh while passing all constraints.

### 3.2. Turbine Specifications

The platform was designed around the 15-MW reference turbine, a theoretical turbine developed by the National Renewable Energy Laboratory (NREL), the Technical University of Denmark (DTU), and the University of Maine. This turbine was developed as a conservative estimate of actual industry capabilities. For example, the 12-MW GE Haliade-X turbine was launched in 2021, and so the IEA 15-MW turbine was developed to represent the near future of the industry [15], making it an appropriate choice for development of a novel platform design. This section details the relevant properties of the turbine required for the optimization. More details of its performance can be found in [15], the detailed mass information for the floating platform version can be found in [16], and a CAD file and other specifications can be found in [23].

The specifications of the IEA 15-MW turbine are shown in Table 14.

**Table 14.** IEA 15-MW turbine specifications.

| Feature | Value |
|---|---|
| *Generator* | |
| Rated power (MW) | 15 |
| Power control strategy | Variable speed, collective pitch |
| Rotor diameter (m) | 240 |
| Hub height (m) | 150 |
| Cut-in wind speed (m/s) | 3 |
| Rated wind speed (m/s) | 10.59 |
| Cut-out wind speed (m/s) | 25 |
| Range of rotational speed (RPM) | 5–7.56 |
| *Blade* | |
| Maximum tip speed (m/s) | 95 |
| Swept area (m$^2$) | 45,000 |
| *Turbine component masses* | |
| Nacelle (t) | 507.3 |
| Hub (t) | 190.0 |
| Yaw bearing (t) | 100.0 |
| Blade x3 (t) | 194.1 |
| Total (t) | 991.4 |

Table 15 provides the quasi-static power coefficient, thrust coefficient, and thrust force for the turbine, including the turbine aerodynamics and control systems.

**Table 15.** Turbine quasi-static characteristics.

| Wind Speed (m/s) | Power (MW) | $C_P$ | Thrust (MN) | $C_T$ |
|---|---|---|---|---|
| 3 | 0.07 | 0.10 | 0.59 | 0.82 |
| 4 | 3.71 | 0.36 | 0.74 | 0.81 |
| 5 | 2.72 | 0.44 | 0.95 | 0.82 |
| 6 | 1.19 | 0.48 | 1.21 | 0.83 |
| 7 | 4.34 | 0.49 | 1.46 | 0.81 |
| 8 | 6.48 | 0.49 | 1.79 | 0.80 |
| 9 | 9.23 | 0.49 | 2.15 | 0.80 |
| 10.59 [1] | 15.0 | 0.49 | 2.73 | 0.77 |
| 11 | 15.0 | 0.44 | 2.38 | 0.61 |
| 12 | 15.0 | 0.34 | 2.05 | 0.43 |
| 13 | 15.0 | 0.26 | 1.86 | 0.32 |
| 14 | 15.0 | 0.21 | 1.72 | 0.25 |
| 15 | 15.0 | 0.17 | 1.62 | 0.20 |
| 16 | 15.0 | 0.15 | 1.54 | 0.17 |
| 17 | 15.0 | 0.12 | 1.47 | 0.14 |
| 18 | 15.0 | 0.10 | 1.41 | 0.12 |
| 19 | 15.0 | 0.09 | 1.36 | 0.16 |
| 20 | 15.0 | 0.07 | 1.31 | 0.09 |
| 21 | 15.0 | 0.06 | 1.28 | 0.08 |
| 22 | 15.0 | 0.05 | 1.25 | 0.07 |
| 23 | 15.0 | 0.05 | 1.21 | 0.06 |
| 24 | 15.0 | 0.04 | 1.19 | 0.05 |
| 25 | 15.0 | 0.04 | 1.17 | 0.05 |

[1] Rated wind speed.

The peak thrust value provided at the rated wind speed was used in the calculation of $g_4$, the HDF constraint for when the linear hydrostatics were violated. The mass and geometry presented above give an overview of the what was needed for the calculate masses, COGs, and moments in the HDF. More detailed specifications were obtained from the OpenFAST input files found on GITHUB [23].

*3.3. Wind and Wave Conditions*

The wind and wave conditions were developed with data for a project site in state waters approximately 4 km south of Monhegan Island, Maine, USA. This site is representative of the typical conditions found off the northeastern coast of the United States, and it was deemed appropriate for offshore wind turbine systems under the ARPA-e ATLANTIS program [6]. Water depths in the area are variable, ranging from 60 to 110 m. The site is approximately 1.78 km by 3.38 km and is bounded at the southern edge by a 4.83-km line indicating the extent of Maine state waters. The boundary coordinates are as follows: northern = 43°43′18.231″; eastern = 69°20′16.759″; southern = 43°42′15.436″; and western = 69°17′36.544″. A map of the site is shown in Figure 14.

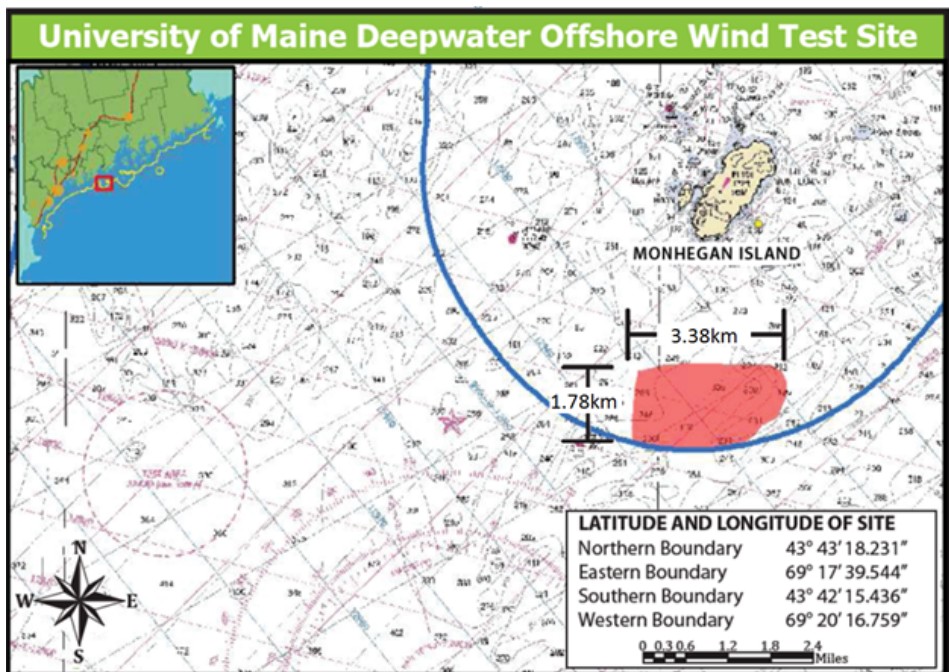

**Figure 14.** Map of the project site's location.

The design conditions were based on approximately 12 years of oceanographic buoy data collected by the UMaine Physical Oceanography Group (PhOG) within the School of Marine Sciences less than 2.5 km from the test site. For more information on the data collection process or to download the data, refer to the UMaine buoy website [24].

The design conditions presented within this work were derived with the use of data collected from three (3) metocean buoys. The majority of the data presented here was derived from 13 years of Buoy E01 measurements. The buoy collected the following data: significant wave heights and peak periods, 8-minute average and 3-second gust wind speeds and directions, sea and air temperatures, current speed and direction from 2 m to 62 m below sea level, and air pressure. However, the E01 system did not record the mean wave direction and as such was supplemented with 2 years of data from Buoy E02 over two deployments in 2011 and 2015 at the test site. Additionally, the wave spectrum parameters for the region were derived from 10 years of data collected from NOAA Station 44007.

- UMaine PhOG designation: E01
  NOAA buoy designation: station 44032;
  Deployment location: 43°42′94″ N, 69°21′32″ W;
  Data range used: 9 July 2001–12 September 2014;
  Data types used: significant wave height, peak wave period, wind speed and direction, and current speed and direction.
- UMaine PhOG designation: E02
  NOAA buoy designation: N/A;
  Deployment location: 43°42′39″ N, 69°19′18″ W;
  Data range used: 11 August 2010–7 October 2011 and 14 November 2014–17 September 2015;
  Data types used: significant wave height and mean wave direction.
- UMaine PhOG designation: N/A
  NOAA buoy designation: station 44007;
  Deployment location: 43°31′30″ N, 70°8′26″ W;
  Data range used: 1 January 2007–20 June 2017;
  Data types used: wave spectral parameters.

Analysis of the data presented here was completed following the guidelines of the International Standard IEC 61400-1 [25] and IEC 61400-3 [26] (Wind Turbines: Design re-

quirements and design requirements for offshore wind turbines). The resulting data points required to generate the design load cases are shown in Table 16. Next to each parameter is the citation used to calculate each value. Note that for the individual extreme wave heights, the significant wave height values were from [27], with their heights multiplied by 1.86 per guidance from [26]. The extreme sea currents at varying depths were obtained from peaks over threshold analysis from Buoy EO1 with a generalized pareto extreme value distribution.

**Table 16.** Summary of environmental design parameters.

| Wind Design Parameters | Value |
| --- | --- |
| Annual average wind speed at 100 m (m/s) [28] | 8.75 |
| Extreme 10-min average 1-year wind speed at 4 m (m/s) [27] | 18.4 |
| Extreme 10-min average 10-year wind speed at 4 m (m/s) [27] | 21.8 |
| Extreme 10-min average 50-year wind speed at 4 m (m/s) [27] | 24.1 |
| Extreme 10-min average 500-year wind speed at 4 m (m/s) [27] | 26.7 |
| Normal wind shear power law exponent per ABS [19] | 0.14 |
| Extreme wind shear power law exponent per ABS [19] | 0.26 |
| **Metocean and Site Design Parameters** | **Value** |
| 1-year significant wave height (m) [27] | 6.4 |
| 10-year significant wave height (m) [27] | 8.5 |
| 50-year significant wave height (m) [27] | 9.8 |
| 500-year significant wave height (m) [27] | 11.5 |
| Mean peak period associated with 1-year sig wave height (s) [27] | 11.7 |
| Mean peak period associated with 10-year sig wave height (s) [27] | 13.3 |
| Mean peak period associated with 50-year sig wave height (s) [27] | 14.2 |
| Mean peak period associated with 500-year sig wave height (s) [27] | 15.0 |
| 1-year individual extreme wave height (m) [27] | 11.9 |
| 10-year individual extreme wave height (m) [27] | 15.8 |
| 50-year individual extreme wave height (m) [27] | 18.2 |
| 500-year individual extreme wave height (m) [27] | 23.0 |
| Extreme 1-year sea current at depths 2 m/10 m/30 m/62 m (cm/s) [24] | 77/63/48/45 |
| Extreme 1-year sea current at depths 2 m/10 m/30 m/62 m (cm/s) [24] | 89/79/67/67 |
| Extreme 50-year sea current at depths 2 m/10 m/30 m/62 m (cm/s) [24] | 105/88/81/88 |
| Extreme 500-year sea current at depths 2 m/10 m/30 m/62 m (cm/s) [24] | 127/99/104/129 |

Taking the data points from Table 16, the design load cases used in the optimization were developed, and they are summarized in Table 17. As detailed in the Materials and Methods section of this report, a subset of the full DLCs was used to save computation time based on those conditions which caused constraint failures. $H_s$ is the significant wave height, $T_p$ is the peak period, and $\gamma$ refers to the spectral shape parameter for the JONSWAP. Each case was considered with wind, wave and current headings of 90° from true north to minimize the simulation cases, which was aligned with the legs. The wind speeds are listed at hub height, and the current speeds are at a 2-m depth.

**Table 17.** Environmental conditions for DLCs included in simulation.

| DLC | Wind Speed (m/s) | $H_s$ (m) | $T_p$ (s) | fl | Current Speed (m/s) |
|-----|-----|-----|-----|-----|-----|
| 1.1 | 10 | 1.03 | 7.12 | 1.5 | 0.158 |
| 1.1 | 24 | 3.07 | 9.01 | 1.8 | 0.307 |
| 1.6 | 10 | 8.1 | 12.8 | 2.75 | 0.158 |
| 1.6 | 12 | 8.5 | 13.1 | 2.75 | 0.163 |
| 1.6 | 14 | 8.5 | 13.1 | 2.75 | 0.174 |
| 1.6 | 16 | 9.8 | 14.1 | 2.75 | 0.190 |
| 1.6 | 18 | 9.8 | 14.1 | 2.75 | 0.211 |
| 1.6 | 20 | 9.8 | 14.1 | 2.75 | 0.238 |
| 1.6 | 22 | 9.8 | 14.1 | 2.75 | 0.270 |
| 1.6 | 24 | 9.8 | 14.1 | 2.75 | 0.307 |
| 6.1 | 58.7 | 10.7 | 14.2 | 2.75 | 1.05 |

*3.4. Genetic Algorithm Specifications and Convergence*

The objective and constraint functions were written for a genetic algorithm MATLAB code as used in [14]. The input parameters determining the convergence criteria, crossover, mutation, and niching behavior are listed in Table 18. Only the maximum generations, population, and number of genes were tuned from a set of values designed to work for most problems. Specifically, with 6 input variables, the number of genes was also 6, and the number of individuals in the population was set at 120, or 20 times the number of genes. The maximum number of generations was set at 100.

**Table 18.** Genetic algorithm specifications.

| Parameter | Value |
|-----|-----|
| Maximum generations | 100 |
| Population number | 120 |
| Number of genes | 6 |
| Elite parameter | 1 |
| Best parameter | 1 |
| Probability of crossover | 0.9 |
| Probability of SBX crossover | 0.5 |
| Crossover strength parameter | 1 |
| Probability of mutation | 0.02 |
| Probability of PBM operation | 0.5 |
| Mutation strength parameter | 100 |
| Maximum allowable niching distance | 0.1 |
| Individuals checked during niching parameter | 0.25 |
| Drop parameter | 0.5 |
| Dyn parameter | 0.001 |

To check that the genetic algorithm was not stuck in a local minima, multiple runs were performed. By ensuring that the values of the genes for each run were close to each other, it was concluded that the solution was adequately converged. Table 19 lists the values between runs and their differences, expressed as percentages.

**Table 19.** Converged values for different optimizer runs.

| Variable | Optimizer Run 1 | Optimizer Run 2 | Percent Difference |
|-----|-----|-----|-----|
| $r$ (m) | 37.58 | 37.89 | 0.83 |
| $w$ (m) | 15.53 | 14.86 | 4.37 |
| $d$ (m) | 12.50 | 12.33 | 1.37 |
| $h_p$ (m) | 6.33 | 6.79 | 6.98 |
| $f$ (m) | 6.14 | 6.65 | 7.92 |
| $a$ | 1.90 | 1.99 | 4.51 |

The standard deviation among the population in the last generation was also examined. In the final generation, there should be a low standard deviation, indicating a limited spread of designs around the best individual. For example, Table 20 shows the standard deviations for one of the optimization runs.

**Table 20.** Standard deviation for the 100th generation.

| Variable | Converged Value | Standard Deviation |
|:---:|:---:|:---:|
| $r$ (m) | 37.58 | 0.535 |
| $w$ (m) | 15.53 | 0.507 |
| $d$ (m) | 12.50 | 0.295 |
| $h_p$ (m) | 6.33 | 0.117 |
| $f$ (m) | 6.14 | 1.69 |
| $a$ | 1.90 | 0.069 |

To further illustrate the convergence of the optimizer, the histograms of the population were created at different generations. At the start of the optimization run, the population followed a random distribution across the range of possible input variable points, as shown in Figure 15. After 50 generations, the genetic algorithm began to find favorable designs, and thus the population followed a distribution centered around specific gene values, as shown in Figure 16. After 100 generations, the standard deviation of the designs was very low, so almost all the design points were tightly clustered around the best values, as shown in Figure 17.

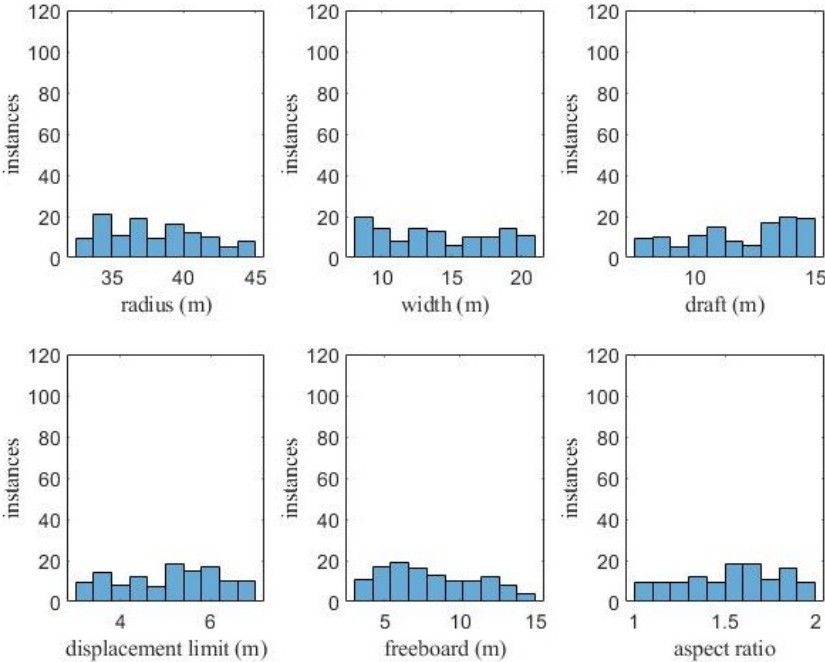

**Figure 15.** Population histogram for the first generation.

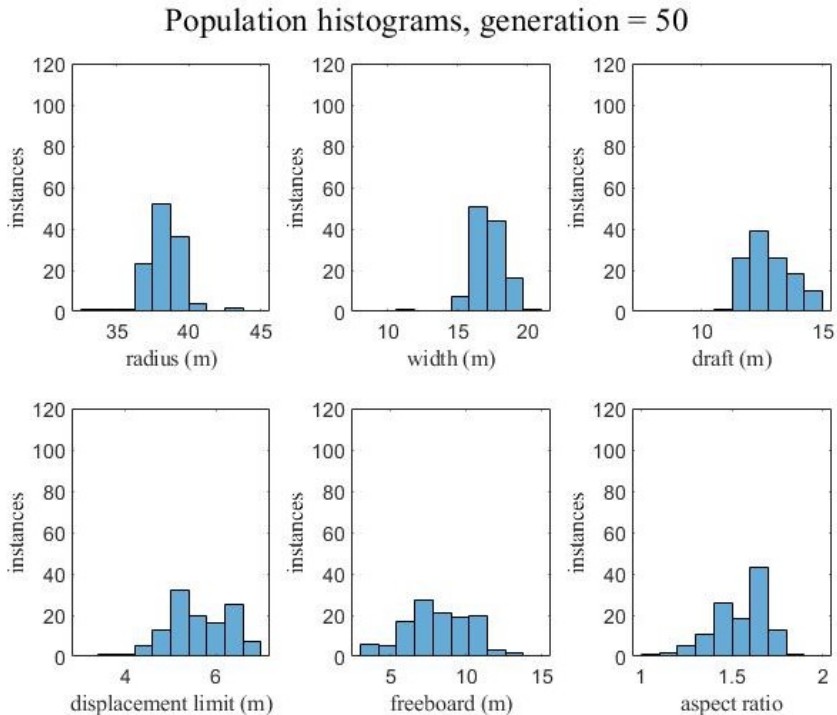

**Figure 16.** Population histogram for the 50th generation.

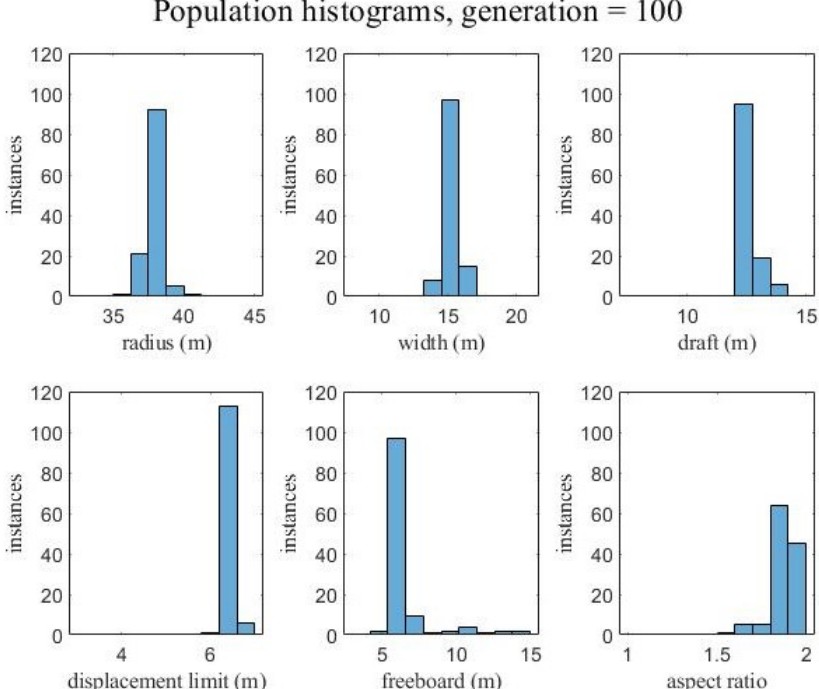

**Figure 17.** Population histogram for the 100th generation.

Another way of confirming the optimizer resulting in the quasi-optimum solution is to plot the surfaces of the input variables against the LCOE with the constraint values overlayed in color. For example, plotting the radius and width of the platform against the LCOE yielded Figure 18. Here, the darkest blue indicates designs that passed all constraints, with shading of yellow indicating constraint failure. Since the staged constraint approach yielded some designs with very high constraint values relative to the designs that just barely failed the constraints, the constraints were normalized to better show the resolution

of shading on the plot. The red point shows the optimized design, which is just at the edge of failing the constraints, and also at the minimum possible LCOE that still passes the constraints. This indicates the best possible design for the problem posed.

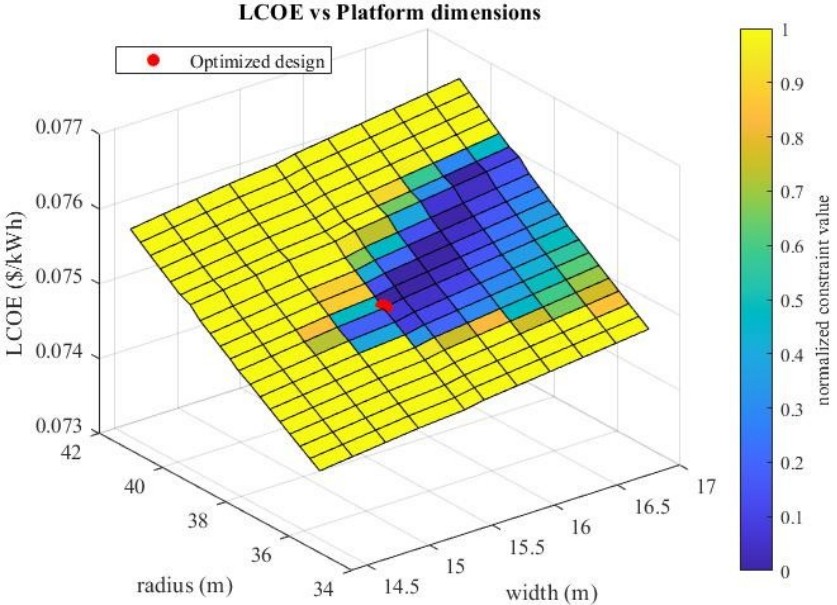

**Figure 18.** Surface plot of LCOE vs. radius and width with constraint values overlayed on the surface.

### 3.5. Optimized Platform Design

This section presents information about the overall dimensions, masses, and COGs of the optimized platform and are compared to a baseline design. The baseline design was initially developed to demonstrate the potential for the damper concept, and it is provided to demonstrate the changes in properties when the system was optimized. Note that the baseline design was designed before the FDF constraints were designed and did not pass all the motion constraints. Therefore, this explains why the change in LCOE was not more dramatic. Additionally, the FDF inputs and dynamic performance as they relate to the constraints are presented.

### 3.5.1. Hydrostatic Specifications

The input variables values for the optimized platform are listed in Table 21. These variables correspond to those labeled in Figure 6. The optimized values were found to minimize the LCOE while passing all the constraints, and more details on the convergence criteria are provided in Genetic Algorithm Specifications and Convergence.

**Table 21.** Input variable converged values.

| Variable | Optimized | Baseline | Percent Change |
|---|---|---|---|
| $r$ (m) | 37.58 | 43.50 | −13.61 |
| $w$ (m) | 15.53 | 11.00 | 41.18 |
| $d$ (m) | 12.50 | 10.50 | 19.05 |
| $h_p$ (m) | 6.33 | * | * |
| $f$ (m) | 6.14 | 8.00 | −20.88 |
| $a$ | 1.90 | * | * |

* The starred values were not compared because the baseline system was not designed around the present damper design.

Overall, the legs were made shorter, and the freeboard was reduced, but the widths of the legs and the draft were increased to allow for a greater ballast mass.

The general properties for the converged platform are listed in Table 22. This table also compares the parameters for the baseline platform. The values for the displacement, COGs, and inertias in Table 22 include the mass of the IEA 15-MW turbine.

Observing the changes between the baseline system and the optimized system allows for some conclusions on the characteristics favored by the optimizer to be formed. The ballast mass was more than twice the mass of the hull concrete mass because the dampers were more effective with more ballast mass, and the relatively lightweight hull required a significant amount of ballast to float at the specified draft. Although the waterplane area increased the heave and pitch stiffnesses, this was countered by the increase in mass from the ballast, resulting in lengthened heave and pitch natural periods. The heave natural period stayed within the wave period avoidance range, and the pitch natural period was outside of the typically avoided 5–20 s.

**Table 22.** Mass and hydrostatic properties for the optimized platform.

| Parameter | Optimized | Baseline | Percent Change |
|---|---|---|---|
| Hull displacement ($m^3$) | 26,170 | 18,827 | 39.00 |
| Waterplane area ($m^2$) | 2093 | 1790 | 16.93 |
| Hull concrete mass (t) | 7084 | 9382 | −24.59 |
| Ballast mass, fluid (t) | 15,850 | 6853 | 131.3 |
| TMD equipment steel mass (t) | 821.9 | * | * |
| Vertical COG from SWL (m) | 6.701 | 10.82 | −38.07 |
| Vertical COB from SWL (m) | −6.251 | −5.25 | 19.07 |
| Roll inertia about COG ($kg \cdot m^2$) | $3.399 \times 10^{10}$ | $2.924 \times 10^{10}$ | 16.24 |
| Pitch inertia about COG ($kg \cdot m^2$) | $3.410 \times 10^{10}$ | $2.924 \times 10^{10}$ | 16.62 |
| Yaw inertia about COG ($kg \cdot m^2$) | $1.464 \times 10^{10}$ | $1.027 \times 10^{10}$ | 42.55 |
| KG (m) | 19.20 | 21.32 | −9.94 |
| KB (m) | 6.25 | 5.25 | 19.05 |
| BM (m) | 21.70 | 32.51 | −33.25 |
| GM (m) | 8.75 | 16.44 | −46.78 |
| Heave natural period (s) | 11.38 | 9.81 | 16.00 |
| Pitch natural period (s) | 27.15 | 21.61 | 25.64 |

* The TMD equipment steel mass was not considered in the baseline design.

The FDF assumes the pitch stiffness is constant. However, the stiffness varies with the motion of the ballast water because of the influence of the vertical center of gravity on the righting moment. An estimate of the range of possible values for the pitch stiffness is shown in Table 23. The effects of the changing stiffness were not considered, and this is a limitation of the model, but it is not one with a significant change in the results.

**Table 23.** Change in pitch stiffness with TMD motion.

| TMD Position | Pitch Stiffness (Nm/rad) | Percent Change vs. Resting |
|---|---|---|
| Up limit | $1.84 \times 10^9$ | −17.61 |
| Resting | $2.23 \times 10^9$ | 0 |
| Down limit | $2.63 \times 10^9$ | 17.61 |

The platform with the IEA 15-MW turbine is shown in Figure 19. This view shows the hub height, rotor diameter, draft, and freeboard. All platform designs maintained the 150-m hub height, so based on the value of the freeboard, the height of the tower interface changed to maintain the hub height. The mooring system, which was assumed to have a constant pretension, is not shown.

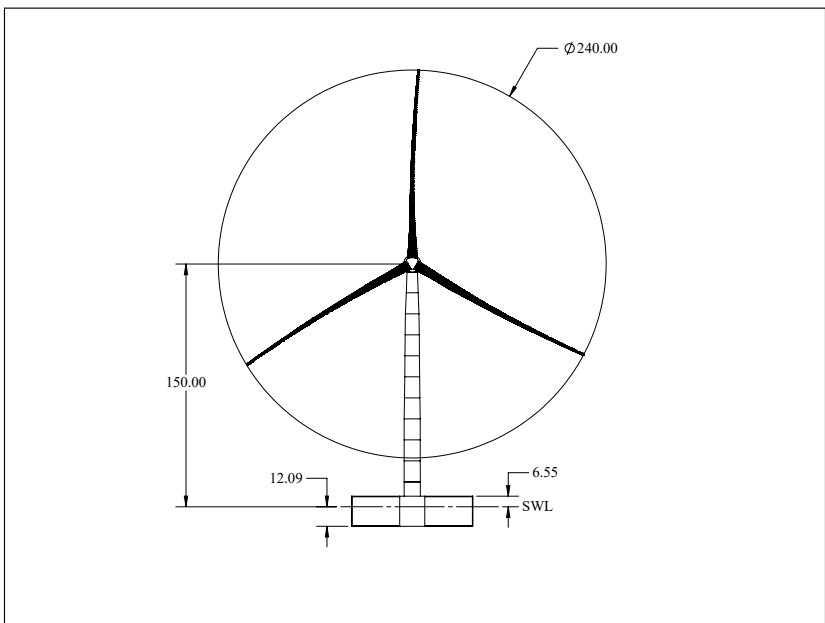

**Figure 19.** Drawing of the platform with IEA 15-MW turbine.

A view of the platform showing the outer dimensions is shown in Figure 20.

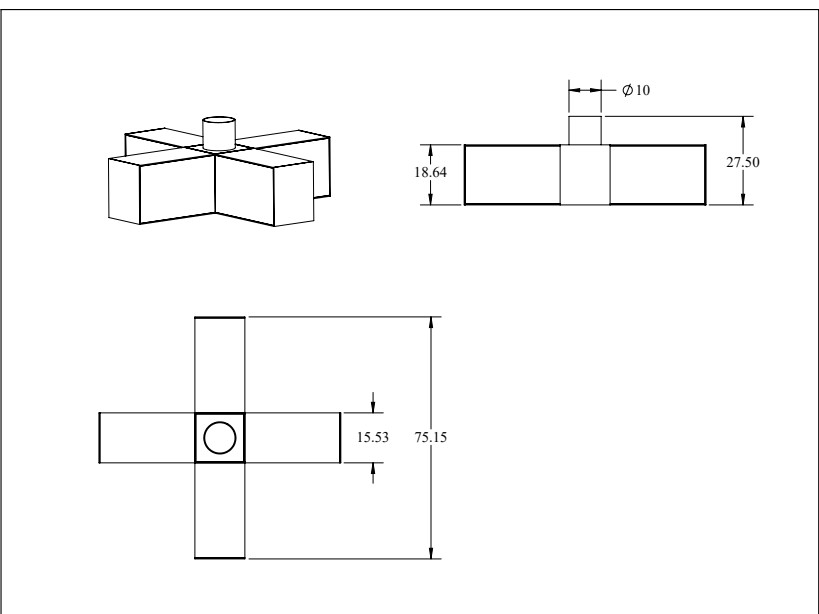

**Figure 20.** Drawing of the hull.

The internals of the platform are shown in Figure 21. Noting the thin wall thickness relative to the scale of the drawing, the dimensioning in this view is based on the internal distances versus the external distances shown in Figure 20. This view shows the wall between the ballast chamber and the keystone with very little vacancy between them. This is because the optimizer favored the aspect ratio to produce long ballast chambers relative to the width. The mass, COG, and moments of inertia of this component were included in the optimizer. However, after the final design, the mass from this component would be replaced by ballast water. As noted in the Materials and Methods section, the line of action of the dampers was assumed to be in the center of the ballast chambers in the plan.

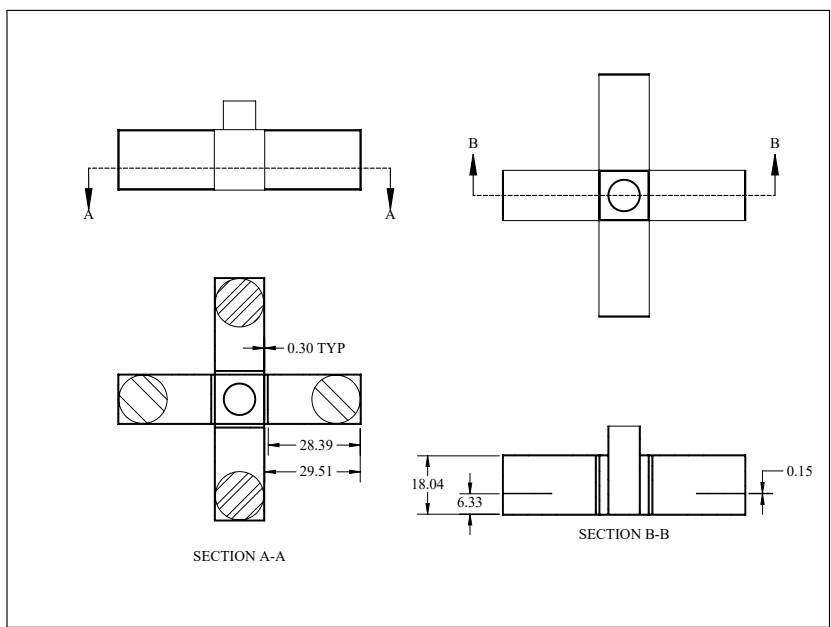

**Figure 21.** Drawing of the internal geometry of the platform.

3.5.2. Frequency Domain Inputs and Dynamic Performance

The hydrostatic function took the input variables and generated inputs for the frequency domain function shown in Table 24. The hydrostatic and frequency domain constraints were all zero for the optimized platform.

**Table 24.** Frequency domain inputs.

| Matlab Variable | Value |
|---|---|
| $L_{wz}$ (m) | $-6.701$ |
| $I_s$ (kg $\cdot$ m$^3$) | $3.410 \times 10^{10}$ |
| $K_{11}$ (N/m) | $6.360 \times 10^4$ |
| $K_{33}$ (N/m) | $2.104 \times 10^7$ |
| $z_{cg,tower}$ (m) | 49.31 |
| $M_{tower}$ (kg) | 1,263,000 |
| $z_{cg,hull}$ (m) | $-9.636$ |
| $M_{hull}$ (kg) | $7.084 \times 10^6$ |
| $z_{cg,RNA}$ (m) | 142.2 |
| $M_{RNA}$ (kg) | $9.914 \times 10^5$ |
| $Mp_{total}$ (kg) | $1.585 \times 10^7$ |
| $Mp_{xcg}$ (kg) | 23.08 |
| $Mp_{zcg}$ (kg) | $-8.093$ |
| $L_{tbz}$ (m) | 8.299 |

The controller scheduling described in Chapter 1 resulted in a period of 19.47 s. The best damping ratio and platform motions are shown in Table 25. The variables $r_6$, $r_7$, $r_8$, and $r_9$ are the platform motions described in Chapter 1 (the RNA horizontal max acceleration, the RNA vertical max acceleration, the max pitch angle, and the max TMD displacement, respectively), and *Twbsmt* is the tower base moment in kN $\cdot$ m. The constraint for $r_7$, the vertical RNA acceleration (limited to 2.00 m/s$^2$) was just barely passed. Additionally, although further investigation would be required, it is important to note that the damping ratio stayed relatively constant for DLCs 1.6 and 6.1, which were the limiting motion cases. It is likely that in the real design, a constant damping ratio tailored for the limiting motion cases would suffice.

**Table 25.** Control scheduling and platform motions.

| DLC/Wind Speed | $\imath$ | $r_6$ (m/s$^2$) | $r_7$ (m/s$^2$) | $r_8$ ($°$) | Twbsmt (kN · m) | $r_9$ (m) |
|---|---|---|---|---|---|---|
| DLC 1.1/10 m/s | 3 | 0.390 | 0.175 | 7.139 | $4.46 \times 10^5$ | 0.127 |
| DLC 1.1/24 m/s | 1 | 0.731 | 0.640 | 3.285 | $1.96 \times 10^5$ | 1.146 |
| DLC 1.6/10 m/s | 0.7 | 1.313 | 1.630 | 8.570 | $6.12 \times 10^5$ | 5.081 |
| DLC 1.6/12 m/s | 0.7 | 1.262 | 1.673 | 8.227 | $5.77 \times 10^5$ | 5.359 |
| DLC 1.6/14 m/s | 0.7 | 1.504 | 1.680 | 7.332 | $5.34 \times 10^5$ | 5.359 |
| DLC 1.6/16 m/s | 0.9 | 1.561 | 1.847 | 5.151 | $4.15 \times 10^5$ | 5.339 |
| DLC 1.6/18 m/s | 0.9 | 1.640 | 1.846 | 4.477 | $4.01 \times 10^5$ | 5.339 |
| DLC 1.6/20 m/s | 0.9 | 1.538 | 1.846 | 4.234 | $3.82 \times 10^5$ | 5.339 |
| DLC 1.6/22 m/s | 0.9 | 1.684 | 1.848 | 4.326 | $3.81 \times 10^5$ | 5.339 |
| DLC 1.6/24 m/s | 0.9 | 1.698 | 1.847 | 4.320 | $3.57 \times 10^5$ | 5.339 |
| DLC 6.1/58.7 m/s | 0.9 | 1.415 | 1.999 | $-0.252$ | $7.99 \times 10^4$ | 5.822 |

In summary, of the motions presented for each of the DLC cases from Table 25, the maximum values are listed in Table 26 with the corresponding DLC and wind speeds indicated.

**Table 26.** Maximum platform motions.

| Property | Maximum Value | DLC/Wind Speed |
|---|---|---|
| Horizontal RNA Acceleration (m/s$^2$) | 1.698 | DLC 1.6/24 m/s |
| Vertical RNA Acceleration (m/s$^2$) | 1.999 | DLC 6.1/58.7 m/s |
| Platform Pitch ($°$) | 8.570 | DLC 1.6/10 m/s |
| Tower Base Moment (kN · m) | $6.12 \times 10^5$ | DLC 1.6/10 m/s |
| TMD Displacement (m) | 5.822 | DLC 6.1/58.7 m/s |

To demonstrate the effect of the TMD on the platform, RAOs were produced from the FDF for comparing the motion of the platform with the TMD turned off (TMD motion locked out with infinite damping) to the motion with the TMD on. The TMD period was set to 19.47 s, and the damping ratio was held constant at 0.9, since this value was the most effective one in the majority of the DLCs. The heave RAO is shown in Figure 22, and the pitch RAO is shown in Figure 23.

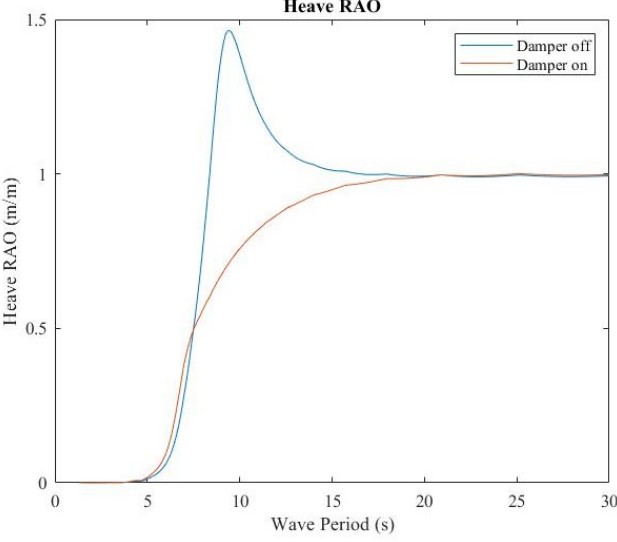

**Figure 22.** RAO comparing the platform heave with the TMD on and off.

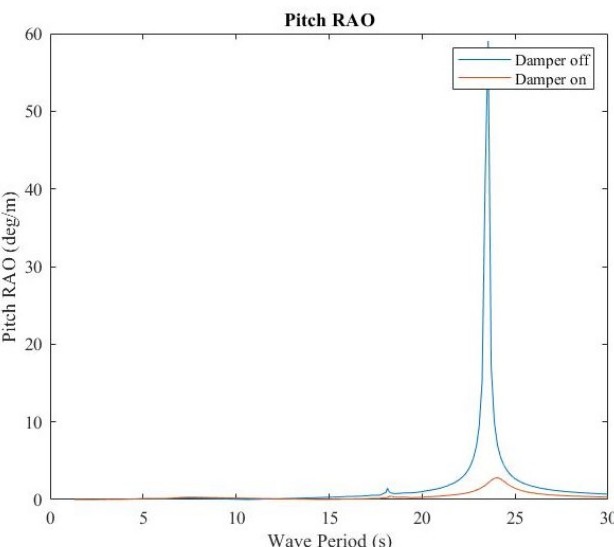

**Figure 23.** RAO comparing the platform pitch with the TMD on and off.

The heave RAO shows the TMD being effective within the wave period avoidance range, with a significant reduction. The large reduction in motion for the pitch RAO shows that without the TMD working, the design would be unsuitable, but the inclusion of the TMD resulted in a significant reduction in platform motion. It is interesting to note that whereas the peak spectral period $T_p$ of the waves was at 12.8 s for DLC 1.6 with a 10-m/s wind speed (triggering the highest pitch angle), the control scheduling for the TMD chose a period of 19.47 s. This fell between the heave and pitch natural periods of 11.38 s and 27.15 s respectively. This suggests that the primary benefit of the TMD is avoidance of the resonant conditions of the platform in storm conditions, as opposed to operational cases.

Providing confidence in the optimization results was the fidelity of the FDF. As reported in [17], OpenFAST, a commonly used time domain solver for floating offshore wind turbines, had good agreement with the frequency domain model used here. Table 27, for example, shows the percent difference in response between the frequency domain and OpenFAST model for the storm case of DLC 6.1. This was for a generic cruciform design and not the optimized design. While the accuracy was sufficient for the optimization, it is important to understand why the differences arose.

**Table 27.** Comparison of frequency domain model and OpenFAST [17].

| Motion | % Diff. Freq. Dom. and OpenFAST |
|---|---|
| RNA horizontal acceleration | 12.2 |
| RNA vertical acceleration | 1.5 |
| Platform pitch | 7.6 |
| Heave | 4.4 |

When examining the RNA vertical and horizontal accelerations, the vertical response stayed close between the frequency domain model and the OpenFAST model, while the horizontal response was underestimated, especially near the peak thrust. The vertical acceleration is primarily a result of the heave motion of the platform, with only secondary effects from the pitch motion [29]. Contrasting this, the horizontal RNA motion is directly a function of both hydrodynamic and aerodynamic loading, the latter of which OpenFAST represents as a nonlinear quantity, whereas the frequency domain model does not.

The pitch angle was also present as a constraint in this optimization, which showed good agreement. The largest discrepancy occurred between 8 and 12 m/s of input wind speed. In this region, the platform is more sensitive to nonlinear turbine controller effects [30], which are captured in the OpenFAST model but not the linearized frequency

domain function, explaining the difference. Overall, the optimization techniques applied had sufficient accuracy and computational efficiency to generate a viable initial design.

## 4. Discussion

An optimization framework for a novel floating platform concept using a TMD was successfully completed, with the result of meeting the desired cost targets with an LCOE of USD 0.0753/kWh and passing the constraints. The overall mass of the platform was 7,905,400 kg, which as a percentage of the equivalent steel mass of the entire system was 15.4%, a significant reduction from existing platform designs. Considering the cost of existing floating offshore wind technologies, meeting the cost targets set by ARPA-E is a significant step toward further development of the concept and toward increasing the viability of the offshore wind resource to power homes. Furthermore, successful execution of the methods proposed in this work indicates the potential for a design methodology shift, where components can be optimally sized for both cost and design constraints simultaneously. Although the final design work has yet to check the strength requirements, make detailed designs of the TMD elements, run the model through a full suite of design load cases, and conduct model testing, the work presented here is a promising step.

Since the post-tensioned concrete hull is significantly lighter than its equivalent mass in steel, the design bypasses one of the primary barriers to offshore wind: the high capital expenditure in materials. In addition to the cost reductions allowed by the cheaper material, this change was allowed by the optimization of the TMD with the platform. Since the platform was designed around the TMD from the start, it could be used to avoid the primary excitation modes. The typical wave period avoidance requirements of offshore platform design were bypassed, significantly decreasing the necessary mass of the platform.

In the analysis of the platform, the genetic algorithm coupled with a unique constraint handling technique provided insight on floating offshore turbine platform design techniques. The majority of a typical design process was automated in the form of MATLAB functions to handle the initial hydrostatic calculations and dynamic response predictions. Many prior works have optimized only parts of the design, such as a damping element or the outer dimensions of a hull. However, by automating the hydrostatic and dynamic calculations to produce the necessary constraints, the optimizer was able to find the best TMD element together with the hull, ultimately producing a less expensive design. Crucially, with the use of the staged constraint handling technique and the frequency domain function, the optimization could find a solution within a relatively short amount of time.

The optimization handled a significant portion of the design, but the final design work remains before the platform is ready for a model test and further development. Specifically, three important areas of future work were not covered in this optimization: detailed structural analysis, the full set of design load cases required by the IEC, and detailed design of the TMD elements.

There were no structural load related constraints included in the optimization. Instead, a conservative estimate of the wall thickness, kept uniform throughout the hull, was used based on a preliminary design. A future version of the optimizer could include wall thickness as an input variable and simple analytical expressions to calculate the constraints. Optimization of the wall thickness could potentially result in a lighter platform. Additionally, detailed structural calculations must be made with the potential to add local sizing adjustments and reinforcements.

Although every effort was made to identify the limiting design load cases to include in the optimization, the cases included were only a small subset of those required for certification. Upon running time domain simulations of all design load cases, if a case was found that exceeded dynamic constraints, the optimization would need to be rerun with that design load case.

The TMD element used in the optimization was not designed in detail because of project time constraints. As a result, simplifications were made to the model with the expectation that detailed specification would take place in a future design phase.

The method developed in this optimization was a step forward in terms of a platform design with the use of the TMD and a simple post-tensioned concrete hull. The optimization techniques could also be a guide to future work. The MATLAB functions described here were specific to the design of this platform, but as floating offshore wind turbine design techniques advance, a more general optimization tool could be developed for research use with user-defined defined platform concepts.

**Author Contributions:** Conceptualization, A.G., C.A., A.V. and R.K.; Formal analysis, C.A.; Funding acquisition, A.V.; Investigation, W.R., A.G. and C.A.; Methodology, W.R., A.G., C.A. and R.K.; Software, W.R. and C.A.; Supervision, C.A. and A.V.; Writing—original draft, W.R.; Writing—review & editing, A.G. and W.R. All authors have read and agreed to the published version of the manuscript.

**Funding:** This research was funded by the Advanced Research Projects Agency–Energy award number DE-AR0001188.

**Institutional Review Board Statement:** Not applicable.

**Informed Consent Statement:** Not applicable.

**Data Availability Statement:** Wind and wave data for the site off the coast of Maine are available at the cited sources in the Results section. Specifications for the IEA 15-MW reference turbine are publicly available as noted in this article.

**Conflicts of Interest:** The authors declare no conflict of interest.

## Abbreviations

The following abbreviations are used in this manuscript:

| | |
|---|---|
| RNA | Rotor and nacelle assembly |
| LCOE | Levelized cost of energy |
| DOE | Department of Energy |
| ARPA-E | Advanced Research Projects Agency–Energy |
| ATLANTIS | Aerodynamic Turbines Lighter and Afloat with Nautical Technologies and Integrated Servo-control |
| TMD | Tuned mass damper |
| GA | Genetic algorithm |
| HDF | Hydrostatic function |
| FDF | Frequency domain function |
| RSM | Response surface model |
| NREL | National Renewable Energy Laboratory |
| DTU | Technical University of Denmark |

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
