# Peer review of "Optimization of a Lightweight Floating Offshore Wind Turbine with Water Ballast Motion Mitigation Technology"

_2674-032X, doi:10.3390/wind2030029_

Round 1

Reviewer 1 Report

Review Report on Manuscript ID: wind-1820912 with title: "Optimization of a Lightweight Floating Offshore Wind Turbine with Water-Ballast Motion Mitigation Technology"

The current paper presents a recent topic concerning offshore wind turbine progress. The paper falls into the scope of this journal. The organization of this work needs more attention to be suitable as journal work.  The following are the raised comments that are suggested to improve its quality for readers:

1. The article seems as a report on the project as the word (Project) is mentioned more than 15 times. Please, identify correctly.

2. The Abstract section needs more attention with clear findings of this study. 

3. The survey part must be enhanced. 

4. The problem formulation is not clear.

5. The paper presentation is not formal.

6. No comparison with others. 

7. No conclusion section 

8. Most of the references are reports 

Author Response

Dear Reviewer,

Thank you for your insightful comments on our article on a novel floating platform. Investing time to peer review is important but not trivial, so we take your comments, suggestions and concerns very seriously and hope we have addressed them adequately. Each of your comments and our response to them are below.

  1. The article seems as a report on the project as the word (Project) is mentioned more than 15 times. Please, identify correctly.

Uses of the word project were changed to reflect that the journal submission is an article, not a project report.

  1. The Abstract section needs more attention with clear findings of this study. 

The abstract was expanded, highlighting more clearly the features of the platform and design innovations. Additionally, clearer findings of the study were stated. These additions are on lines 8-13.

  1. The survey part must be enhanced. 

A paragraph was added to better summarize the existing literature. It also highlights what the research gap is that this article fulfills. These additions can be found on lines 99-107.

  1. The problem formulation is not clear.

For the less straightforward equations, descriptions were added near the equations in lines 183-186 and 228-232 to clarify the problems being solved.

  1. The paper presentation is not formal.

The article was reviewed and minor wording changes were made, including in regards to point number 1, to improve consistency and formality of the paper.

  1. No comparison with others. 

A table comparing the accuracy of the frequency domain model with the time domain solver OpenFAST was inserted, with three paragraphs explaining the differences between the models. Please see lines 671-690

  1. No conclusion section 

With respectful acknowledgment of the reviewer’s opinion, per MDPI, a conclusion section should only be included if the discussion is excessively long or complicated. It is the opinion of the authors that a separate conclusion adds unnecessary length.

  1. Most of the references are reports 

The authors agree that many of the references are reports, however it is respectfully acknowledged that many of the data for this article come from National Renewable Energy Laboratory (NREL) technical reports, and NREL publishes much of their content in technical report form.

Thank you for your time reviewing these changes.

Sincerely,

William Ramsay

Reviewer 2 Report

This paper presents an optimization  on the floating wind turbine. The focus is interesting in engineering applications. The reviewer has some minor concerns for the improvement of the manuscript.

1. Some key findings or conclusions can be presented in the abstract.

2. Strong nonlinearities exist in the floating wind turbine. How to consider this effect in the frequency-domain optimization? Pls have a comment on this effect and clarify the strategies to involve this effect in optimization.

Author Response

Dear Reviewer,

Thank you for your insightful comments on our article on a novel floating platform. Investing time to peer review is important but not trivial, so we take your comments, suggestions and concerns very seriously and hope we have addressed them adequately. Each of your comments and our response to them are below.

  1. Some key findings or conclusions can be presented in the abstract.

The abstract was expanded, highlighting more clearly the features of the platform and design innovations. Additionally, clearer findings of the study were stated. These additions are on lines 8-13.

  1. Strong nonlinearities exist in the floating wind turbine. How to consider this effect in the frequency-domain optimization? Pls have a comment on this effect and clarify the strategies to involve this effect in optimization.

 A comparison of a common time domain solver (OpenFAST) and the performance of the frequency domain solver was compared to demonstrate the frequency domain solver had sufficient fidelity for optimization. Additionally, any differences between the two solvers were explained in terms of the nonlinear effects captured by OpenFAST where they were linearized in the frequency domain solver. Please see lines 671-690.

Thank you for your time reviewing these changes.

Sincerely,

William Ramsay

Reviewer 3 Report

Reviewer’s report on a manuscript submitted to Journal of Wind (ISSN 2674-032X)

wind-1820912: “Optimization of a Lightweight Floating Offshore Wind Turbine with Water-Ballast Motion Mitigation Technology”

The paper proposed a frequency domain solver with a genetic algorithm (GA) to optimize the design of a novel floating wind turbine concept. The concept relied on liquid ballast mass so as to mitigate motions on a lightweight post-tensioned concrete platform. The paper, in overall, is well written and the topic is interesting. However, the reviewer would like to make the following comments before the paper can proceed further:

-        The abstract must be improved. It can more emphasise on the novelty of the work so as to encourage readers to continue reading the paper.

-    The introduction section is written well. However, the literature review needs to be updated and little more expanded. The main findings of the literature review must be stated clearly and the research gap to be highlighted. The contribution of the paper (the last paragraph of the introduction section) must be expanded too.

-      Some equations in Section 2 must be given a reference as they already exist in the literature.

-      Section 2 is so lengthy, whereas Section 3 is short. Some materials in relation to LCOE calculation can move to Section 3 so as to make Section 2 shorter. Some figures/flowcharts can also move to appendix.

Section 4 is the most interesting part of the paper. However, it lacks rich discussion. The results must be 'discussed' in a more detailed way. Some comparisons between the results and other numerical analysis methods (reported in the literature) would be also insightful.

Author Response

Dear Reviewer,

Thank you for your insightful comments on our article on a novel floating platform. Investing time to peer review is important but not trivial, so we take your comments, suggestions and concerns very seriously and hope we have addressed them adequately. Each of your comments and our response to them are below.

 -  The abstract must be improved. It can more emphasise on the novelty of the work so as to encourage readers to continue reading the paper.

The abstract was expanded, highlighting more clearly the features of the platform and design innovations. Additionally, clearer findings of the study were stated. These additions are on lines 8-13.

-  The introduction section is written well. However, the literature review needs to be updated and little more expanded. The main findings of the literature review must be stated clearly and the research gap to be highlighted. The contribution of the paper (the last paragraph of the introduction section) must be expanded too.

Within the literature review, a paragraph was added summarizing the findings of the literature review, and clearly stating the research gap that needs to be addressed by this article. Additionally, the last paragraph of the introduction was expanded to provide more details on the contributions of this article. Please see lines 90-107 and 153-159.

-   Some equations in Section 2 must be given a reference as they already exist in the literature.

The equations existing in the literature were given a citation right next to them to make it clear where they came from. Please see equations 12 and 13.

-   Section 2 is so lengthy, whereas Section 3 is short. Some materials in relation to LCOE calculation can move to Section 3 so as to make Section 2 shorter. Some figures/flowcharts can also move to appendix.

It was not intended to separate section 2 and section 3, section 3 was meant to be a subheading within section 2. They have now been combined.

- Section 4 is the most interesting part of the paper. However, it lacks rich discussion. The results must be 'discussed' in a more detailed way. Some comparisons between the results and other numerical analysis methods (reported in the literature) would be also insightful.

Section 4 was expanded. First, a comparison of a common time domain solver (OpenFAST) and the performance of the frequency domain solver was compared to demonstrate the frequency domain solver had sufficient fidelity for optimization. Secondly, more discussion on the interaction between the tuned mass damper element and the platform was added to better explain its benefit to the system. Please see lines 664-690.

Thank you for your time reviewing these changes.

Sincerely,

William Ramsay

Round 2

Reviewer 1 Report

No Further comments